# Towards modelling of corrugation ridges at ice-sheet grounding lines

Kelly A. Hogan[1], Katarzyna L.P. Warburton[2,3], Alastair G.C. Graham[4], Jerome A. Neufeld[5,2], Duncan R. Hewitt[6], Julian A. Dowdeswell[7], and Robert D. Larter[1]

[1]British Antarctic Survey, Cambridge, UK
[2]Department of Applied Mathematics and Theoretical Physics, University of Cambridge, Cambridge, UK
[3]Thayer School of Engineering, Dartmouth College, Hanover, NH, USA
[4]College of Marine Science, University of South Florida, St Petersburg, FL, USA
[5]Department of Earth Sciences, University of Cambridge, Cambridge, UK
[6]Department of Mathematics, University College London, London, UK
[7]Scott Polar Research Institute, University of Cambridge, Cambridge, UK

*Correspondence to*: Kelly A. Hogan (kelgan@bas.ac.uk)

**Abstract.** Improvements in the resolution of seafloor mapping techniques have revealed extremely regular, sub-metre scale ridge landforms produced by the tidal flexure of ice-shelf grounding lines as they retreated very rapidly (i.e., at rates of several kilometres per year). Guided by such novel seafloor observations from Thwaites Glacier, West Antarctica, we present three mathematical models for the formation of these corrugation ridges at a tidally migrating grounding line (that is retreating at a constant rate) where each ridge is formed by either constant till flux to the grounding line, till extrusion from the grounding line, or by the resuspension and transport of grains from the grounding-zone bed. We find that both till extrusion (squeezing out till like toothpaste as the ice sheet re-settles on the seafloor), and resuspension and transport of material can qualitatively reproduce regular, delicate ridges at a retreating grounding line as described from seafloor observations. By considering the known properties of subglacial sediments we agree with existing schematic models that the most likely mechanism for ridge formation is till extrusion at each low-tide position, essentially preserving an imprint of the ice-sheet grounding line as it retreated. However, when realistic (shallow) bedslopes are used in the simulations ridges start to overprint one another suggesting that, to preserve the regular ridges that have been observed, grounding line retreat rates (driven by dynamic thinning?) may be even higher than previously thought.

## 1 Introduction

Motion at the grounding line - that is, the junction between ice that is coupled to the bed and freely-floating ice shelves – occurs across a range of timescales and is fundamental to our understanding of marine ice-sheet stability (Thomas, 1979; Schoof, 2007). On short timescales (hours to days), motion is tidally modulated leading to migration over 100s of metres to many kilometres across what is known as the grounding zone (e.g., Rignot et al., 2011; Dawson & Bamber, 2020; Drews et al., 2021; Chen et al., 2023). For longer timescales (multiannual to centennial), migration is forced by climatic factors and ice-dynamic feedbacks leading to long-term trends in ice-sheet advance and retreat. Grounding zones are inherently one of the

most difficult parts of the ice-sheet-ocean system to access making observations of the physical processes that control grounding-line movements particularly challenging. Alternative methods of investigation include mathematical models to simulate grounding-line behaviour (e.g., Gudmundsson et al., 2012; Jamieson et al., 2012; Robel et al., 2014, 2017; Walker et

al., 2013; Tsai & Gudmundsson, 2015; Warburton et al., 2020), laboratory experiments that replicate grounding-line processes (e.g., Pegler and Worster, 2013; Kowal and Worster, 2020), and the study of post-glacial landscapes that represent former grounding zones (e.g., Jakobsson et al., 2012; Simkins et al., 2018; Shackleton et al., 2020). With recent advances in these methods, notably via increasing the resolution at which models are run and post-glacial landscapes (now marine) are observed, it is now possible to test modelled physical processes against observations of incredibly fine-scale glacial landforms produced

at former (marine) grounding zones. The aim of this work is to improve our understanding of the processes acting at ice-sheet grounding lines as they migrate.

One of the most intriguing grounding-zone landforms identified to date are the so-called parallel "ribs" or "rungs" which comprise laterally continuous, low-amplitude ridges (several tens of cm), oriented transverse to ice flow, with relatively uniform spacings (metres to 10s of metres) and morphologies. These subtle sedimentary ridges exhibit a clear 13-15 ridge

periodicity that has led to their interpretation as forming through the tidally modulated motion of ice impacting the sea floor (Graham et al., 2022). Following existing terminology we refer to these landforms as corrugation ridges. Similar ridges are produced in a variety of polar marine settings including iceberg ploughmarks by the forward motion of iceberg keels (Jakobsson et al., 2011), or beneath ice shelves as ice shelf keels periodically ground during unpinning (Graham et al., 2013; see 'Supplementary' in Graham et al., 2022, for a fuller discussion). Recent autonomous underwater vehicle (AUV)

deployments have now mapped these features at high (sub-metre) resolution on the tops of wedges of sediment deposited at the former grounding zones of ice streams emanating from both the Larsen Inlet, eastern Antarctic Peninsula (Dowdeswell et al., 2020) and from Thwaites Glacier, West Antarctica (Graham et al., 2022). The regularity of the spacing between ridges, with a clear periodicity, supports their interpretation as the product of tidal motion as the grounding line lifts and re-settles during an overall pattern of retreat. Because retreat is required to preserve each ridge (Dowdeswell et al., 2020), these

landforms and their spacing allow for the calculation of grounding-line retreat rates on daily timescales as ice receded from what are thought to be stable locations on the constructional sedimentary wedges (cf. Alley et al., 2007). Such direct, quantitative estimates of grounding-line retreat rates provide critical constraints for numerical models seeking to predict patterns and rates of future ice-sheet retreat under a warming climate.

Schematic models of corrugation ridge formation developed so far are conceptual and favour a mechanism by squeezing out

or extrusion of soft sediment from the grounding line as it settles back on the sea floor following the outgoing tide (Dowdeswell et al., 2020; Graham et al., 2022). However, these interpretations are based somewhat on form analogy with other grounding-line landforms such as recessional moraines (which form via push during small grounding-line readvances, e.g., Boulton, 1986), and the mechanism that produces the subtle relief of corrugation ridges with their incredibly regular distribution and geometries has not been explored quantitatively. In this study, we consider corrugation ridge formation as a fluid dynamics

problem with both the subglacial till layer and ocean tides (incoming and outgoing) represented as fluids, overlain by an elastic

(solid) ice sheet. Adapting a 2D flow-line model of grounding-line migration (Warburton et al., 2020), we investigate three different formation mechanisms for corrugation ridges constrained by observations of these landforms in front of Thwaites Glacier. We consider the model results alongside directly observed sea-floor parameters (ridge size, shape, spacing, acoustic character) and with inferred sedimentological properties to test model predictions of corrugation ridge morphology and composition. Finally, we discuss the plausibility of each mechanism based on the known properties of glacial sediments and grounding-zone processes.

## 2. Materials and Methods

### 2.1 Grounding-line migration model

Our aim is to explore formation mechanisms for low-amplitude corrugation ridges produced at the grounding line where one ridge forms during each tidal cycle. In order to preserve each ridge, we require the grounding line to be retreating fast enough that it never overprints its previous low-tide position. Such rapid retreat could be due to an increase in basal melt in the grounding-zone cavity, or ice-flow acceleration and dynamic thinning, or a combination. These are the processes that are driving current mass loss trends for the modern Antarctic and Greenland ice sheets (e.g., Pritchard et al., 2009, 2012; Rignot et al., 2013; Smith et al., 2020). We utilise novel marine geophysical observations of corrugation ridges in front of Thwaites Glacier (Graham et al., 2022; Section 2.2) to inform the model. Thus, in this case we assume a constant retreat rate inferred from the average ridge spacing at Thwaites (6 m) and a retreat rate of 6 m day$^{-1}$ is stipulated in our simulations. We use a tidal component appropriate for the Thwaites area at the present day, as produced by the CATS2008 tide model (Padman et al., 2002; Howard et al., 2019). Tides in the southern Amundsen Sea have a dominant diurnal period with one high tide and one low tide per day; correlations for the Thwaites corrugation ridges indicate that one ridge forms per day, assumed to be at the low-tide position (Graham et al., 2022).

For the model itself, consider an ice sheet with thickness $D(x,t)$ and density $\rho_i$ resting on a bed $b(x)$, so that the bedslope $db/dx$ combined with the ice-thickness gradient $\partial D/\partial x$ produces a subglacial hydraulic pressure gradient $\rho_w g \theta_{eff}$, where $\rho_w$ is the density of water and the effective slope (Figure 1a) is:

$$\theta_{eff} = \frac{db}{dx} - \frac{\rho_i}{\rho_w}\frac{\partial D}{\partial x},\qquad\qquad(1)$$

We use the bedslope at the Thwaites corrugation ridges (typically seaward dipping at 0.5°, corresponding to a value of $db/dx = tan(0.5°)$ of around 0.01) to select an appropriate range of effective slopes for the model. The ice-thickness gradient close to the grounding line at the time of ridge formation is a source of uncertainty in $\theta_{eff}$, with every 10 m reduction in ice thickness per km contributing a further 0.009 to the effective slope (i.e. greater ice-thickness gradients along-flow contribute to an increased effective slope). Assuming that present-day values of the ice-thickness gradient are representative, we show results for $\theta_{eff}$ from 0.02 to 0.04. We also discuss the effect of much lower effective slopes, corresponding to an ice-plain setting c.f. Graham et al. (2022), on ridge formation when describing the model results.

In this geometry (Tsai and Gudmundsson, 2015), a change in ocean height, $\Delta h$, produces a change in equilibrium grounding-line position, $\Delta GL$,

$$\Delta GL = \frac{\Delta h}{\theta_{eff}}, \tag{2}$$

whilst a widespread change in ice thickness, $\Delta D$, produces the required retreat (or advance) of the grounding line by a distance of

$$\Delta GL = \frac{\rho_i}{\rho_w}\frac{\Delta D}{\theta_{eff}}. \tag{3}$$

For the current purpose, we assume over a time interval $t$ a constant thinning rate $r = \partial D/\partial t$ everywhere and an oscillation in ocean height $h_{tide}(t)$ due to tides, resulting in a grounding-line migration pattern that combines tidal migration across the
grounding zone with the mean retreat of that grounding line, given by

$$GL(t) = \frac{1}{\theta_{eff}}\left(\frac{\rho_i}{\rho_w} r \times t + h_{tide}(t)\right). \tag{4}$$

By taking tidal heights ($h_{tide}(t)$) from the CATS2008 tide model for the southern Amundsen Sea area (see middle panels in Figure 3) and using a mean horizontal grounding-line retreat rate ($\rho_i r/\rho_w \theta_{eff}$) from the actual ridge spacing at Thwaites (6 m day$^{-1}$), we only need to estimate $\theta_{eff}$ to be able to model grounding-line migration across the zone of ridge formation.

Below, we describe three mechanisms for corrugation ridge formation that can be tested by the above model. All model parameters, their units and values (if constant) are given in Table 1. The proposed mechanisms are based on physically plausible processes of tidally forced sediment transport and deposition in a grounding-zone tidal cavity (i.e., the space that opens and closes as the grounding line rises and falls with the tides, see Figure 2) including a till extrusion mechanism (Section 2.1.2) that represents the previous descriptive models put forward for corrugation ridge formation (c.f. Dowdeswell et al.,
2020; Graham et al., 2022). Note that in this first study we only consider sediment supply to the grounding line from subglacial till transport or erosion from the existing bed; we do not include the meltout of particles from debris-rich basal ice as a source of material to the grounding zone.

### 2.1.1 Constant till flux and deposition

Based on geophysical observations from extant ice-sheet beds, as well as the landform record of grounding-zone wedges, we
know that deforming subglacial tills are transported across the grounding zone (Alley et al., 1986, 1989; Anandakrishnan et al., 2007). When they are deposited at a stable grounding-line position, advected sediments can form constructional landforms marking that position (e.g., Alley et al., 1989, 2007; Dowdeswell and Fugelli, 2012). A simple mechanism for producing tidally modulated landforms might arise from this concept of constant till deposition over the grounding zone, as the grounding line

moves back and forth across it. The volume of till delivered to any given location will inversely depend on the speed of grounding-line migration across the area, which changes over the tidal cycle.

Suppose that the bed $b(x)$ is initially smooth, and that there is a constant sediment flux $q_{s0}$ (in $m^3\,s^{-1}$) delivered subglacially to the grounding line $GL(t)$ as defined in (4) (Figures 2a, b). Then, the final depth of sediment deposited at the grounding line (in metres) is given by

$$\int_{-\infty}^{\infty} q_{s0} \times \delta\big(x - GL(t)\big)\, dt, \qquad (5)$$

where $\delta$ is the Dirac delta function. For shallow effective slopes, the range over which the grounding line migrates is large, so the speed of migration is slowest (and deposition is greatest) when migration changes direction. This occurs twice per tidal cycle, at high and low tides thus generating two ridges per cycle (Figures 2a, b). In the model for this simple mechanism, the ice sheet does not interact with the ridge after it is formed (e.g., Figure 2b), leading to a complex pattern of superimposed ridges, unless the retreat rate is sufficiently large that the next low-tide position is inland of the high-tide position. We consider this highly unlikely given ice-shelf tidal cavity widths have length-scales of hundreds of metres to kilometres (Rignot et al., 2011; Mohajerani et al., 2021; Chen et al., 2023), far greater than the observed corrugation ridge spacing. More realistically, a grounding line that migrates seaward after high tide would be expected to interact with recently deposited sediments, eroding or reworking them. The high-tide ridge may be flattened or pushed forwards to the low tide grounding-line position as the ice re-settles on the sea floor and migrates downstream resulting in only one ridge per diurnal tide; these processes form the basis of the model in Section 2.1.2.

### 2.1.2 Till extrusion

Here, we consider the movement of till across the grounding zone into a ridge as ice at the grounding line re-settles on the sea floor (after high tide) and then migrates seaward (and pushes sediment) to the low-tide position (Figure 2c, 2d). As a simple idealised model of this process, we assume that the ice has a fixed geometry $d_i$ at its base, and that this shape translates across the bed as the grounding line migrates:

$$d_i(x - GL(t)).$$

Considering the till as a yield-strength fluid (e.g., Boulton, 1987; Boulton & Hindmarsh, 1987) to allow the till to deform, and invoking mass conservation, we assume that any volume of till that the ice displaces as the grounding line migrates seaward is transported to the point where the ice lifts off from the bed. When the grounding line retreats inland, we assume that any available space below the ice becomes filled with water, and the displaced till remains fixed at the low-tide position.

We explore two possible sources of till to form each ridge with this mechanism: a constant till flux as in the previous mechanism ($q_{s0}$; Figure 2c), or mobilisation of existing bed sediments from close to the grounding line due to elastic deformation of the ice, leading to enhanced compression of the sediment by a depth, $d_{comp}$ (Figure 2d; Sayag & Worster 2011). Corrugation ridges will be produced as long as one of these processes occurs.

Whichever the source of till flux to the grounding line, as the tide is coming in a new layer of till is left over the bed (as in the previous mechanism). This layer can then be pushed forwards into a ridge when the tide goes out (Figure 2c). Thus, the total flux of till towards the grounding line is no longer constant but is given by

$$q_s = q_{s0} + \frac{d}{dt}(d_s - d_i). \tag{6}$$

   For simplicity, since the spacing and volume of sediment in the ribs will not be sensitive to the exact shape taken for the ice
lid, we use the steady shape of an ice sheet in hydrostatic balance with the ocean, resting on the till up to the grounding line at which point the cavity opens at an angle α, and (if $d_{comp} > 0$) include a highly simplified form of compression near the grounding line, so that:

$$d_i = -\alpha * \min\left(\frac{d_{comp}}{\alpha}, x\right) \qquad \text{or} \qquad d_i = 0 \text{ if } x > 2\frac{d_{comp}}{\alpha}. \tag{7}$$

   Note that the cavity angle, α, will be the same as $\theta_{\text{eff}}$ if the ice sheet-ice shelf system is in hydrostatic equilibrium, which we
shall assume in the results shown. As a result, the exact shape of the corrugation ridges this model produces closely reflects the choice of basal ice topography, but as we have not included any viscous or erosional mechanism that would smooth the shape of the ridges after their formation we do not expect to be able to compare the detailed ridge morphology (modelled) with observations to evaluate this mechanism.

   In general, the total rib volume is given by:

$V_{rib} = q_{s0} \times t_{total} + d_{comp} \times (\textit{distance between high tide positions}),$

where $t_{total}$ is the time between high tides. We present results for the two limiting cases, either when $d_{comp} = 0$ (all till is sourced from constant flux) or when $q_{s0} = 0$ (all till is sourced from compression at the grounding line).

### 2.1.3 Sediment resuspension in tidal cavities

   A third mechanism for ridge formation involves the erosion and deposition of grains as water enters and exits the ice-shelf
cavity with the tides. As described in Warburton et al. (2020), when the grounding line migrates downstream, water that was brought in by the incoming tide must drain out again. This flux of water could erode sediments from the water-saturated till that is exposed to the ocean at high tide, and deposit it at the low tide grounding-line position once the outgoing tidal flow ceases (Figures 2e, f). This mechanism is somewhat based on the concepts of tidal pumping in the ice-shelf cavity and grounding-zone estuaries, both of which have been put forward as processes capable of eroding the grounding-zone
sedimentary bed (Powell, 1990; Horgan et al., 2013).

   As before, we start from a smooth bed $b(x)$ and deposit a sediment flux $q_s$ at the low-tide grounding line. However instead of a constant flux, we now take $q_s$ to be made of the sediment eroded by the draining water, which is removed at a rate $Q$ from the region between the high tide grounding-line position and the current position and is given by

$$q_s(t) = Q \times \left( \max_{T<t} GL(T) - GL(t) \right). \tag{8}$$

Following the calculation in Warburton et al. (2020), the shear stress, $\tau$, exerted on the bed in the draining region is given by

$$\tau = 2.34(\mu U)^{1/4} B^{1/8} (\rho_w g)^{5/8} \theta_{eff}, \tag{9}$$

where $U = dh_{tide}/dt$ is the speed at which the ocean height lowers, and B is the bending stiffness of the ice sheet as an elastic beam, with flow in the draining region assumed to remain laminar throughout. For physical values of these parameters, the shear stress can reach up to a maximum of approximately 1 N m$^{-2}$. For simplicity, we take the erosion rate $Q$ in the draining

region to be constant, although since $U$ varies over the tidal cycle a more detailed model could include the effect of variable $Q$.

## 2.2 Thwaites marine geophysical datasets

High-resolution multibeam bathymetry and side-scan sonar data were acquired by an autonomous underwater vehicle (AUV) for a 13 km$^2$ area about 3 km offshore from the eastern part of the Thwaites Glacier ice shelf during cruise NBP19-02 of the

*RV Nathaniel B. Palmer* in 2019 (Figure 1b). AUV multibeam bathymetry were cleaned and gridded with horizontal cell sizes of 0.7 and 1.5 m; bottom detect (vertical) resolution is better than 0.02 m. The processed side-scan sonar data, which essentially images sea floor reflectivity strength as acoustic backscatter, is dependent on sediment-grain size, substrate hardness, bed roughness as well as acoustic scattering and incidence, has square 0.05-m pixels. Glacial landform evidence, including subglacial lineations and grounding-zone wedges, confirms that the area surveyed was a former grounding zone for Thwaites

Glacier located on a seafloor high that the glacier was pinned on during its overall retreat (Hogan et al., 2020; Graham et al., 2022). Full details of the AUV datasets at Thwaites Glacier and interpretation of the glacial landforms there are given in Graham et al. (2022).

For this study, we focus on the corrugation ridges from the Thwaites AUV dataset (Figures 1b, S1a). We analysed additional bathymetric profiles across the corrugation ridges to assess ridge morphology in terms of symmetry, lateral continuity, and

ridge shape, and we used the side-scan imagery to inform about acoustic character of the ridges and inter-ridge areas (see Table 2, expanded in Table S1). In addition to these tabulated characteristics, we provide a short summary of the ridge observations here for comparison with our model outputs. The longest series of corrugation ridges at Thwaites contains 164 individual ridges on a shallow seaward-dipping (0.5-2° slopes) seafloor with amplitudes of 0.1-0.7 m, spacings of 1.6-10.5 m. The ridges exhibit a clear 13-15 ridge periodicity in both their amplitude and peak-to-peak spacing that matches the modelled spring-to-

neap tidal periodicity of 14.33 days for the area. The largest spacing between ridges occurs when ridge amplitudes are also highest meaning that taller ribs are further apart than smaller ones, and it is assumed that the largest spacings/amplitudes were formed by the largest tides (i.e., when the grounding-zone is widest and the high-to-low tide positions will be farthest apart). Detailed analyses of individual ridge morphometry showed that the ridges are symmetric in form. Furthermore, the side-scan sonar data indicates that acoustic reflectivity does vary between the ridges (higher backscatter strength) and inter-ridge (lower

backscatter strength) areas, making the ridges particularly easy to identify from this type of data. We have no samples with which to ground truth the side-scan data and these variations may be due to subtle changes in sediment hardness, roughness, grain size or illumination of topography (and shadowed areas) during acquisition. The reader is referred to the Graham et al. (2022) paper and its supplementary material for detailed descriptions of the AUV datasets and interpretation of the corrugation ridges (called "ribs" in their paper).

## 3. Results and discussion

### 3.1 Modelled formation of grounding line corrugation ridges

Modelling the constant till flux mechanism (Section 2.1.1) with a retreat rate of 6 m per tide produces a complex pattern of ridges with some with apparent double-peaked (M-shaped) forms and some with multiple peaks (Figure 3a). These forms result from the close proximity of ridges produced at sequential high-tide positions, and also at low-tide positions, yet there

are not consistently 28 ridges per 14-day tidal cycle. Since each pair of ridges formed by a high and low tide can be separated by 100s of metres, and interleaved by ridges produced by other tides, it is difficult to associate the modelled landforms with the tides that formed them (see Supplementary Video S1). The superposition of several ridges forms complex patterns, with large composite ridges evident on a fortnightly cycle, especially on low effective slopes (Figure 3a). The complex nature of these ridge forms and the production of two ridges per tidal cycle mean that there is no clear correlation between ridge height

and spacing ($R^2 = 0.103$; Figure 3b).

For the till extrusion mechanism (Section 2.1.2), one ridge per day is produced at the low tide grounding-line position (Supplementary Video S2). As might be expected, ridges become more closely spaced during neap tides (or just after) when the horizontal migration is lowest and these merge to produce composite forms on lower effective slopes (Figure 3c). This is because we implement a retreat rate of 6 m per day in our models (to preserve individual ridges) so low-tide positions are also

close together when the overall retreat best balances (or counters) the increase in grounding-line migration on outgoing tides (cf. Figure 2b). This "balance spot" appears sometime after neap tides, as the tides transition from neaps to springs (but is balanced by the retreat rate). One way to visualise this is to consider the case where tidal migration of the grounding line is increasing by 6 m per day, and the rate of grounding-line retreat is also 6 m per day, then successive low-tide ridges will form in the same location and produce composite ridges. As the tidal cycle progresses, this is followed by the formation of taller

ridges during larger tides because the volume of till available to be extruded increases during higher tides. As a result, ridge height and spacing are normally correlated in this mechanism ($R^2 = 0.855$; Figure 3d). Outside areas of composite ridge formation, ridge spacing also increases slightly (ridge bases are wider) on lower bedslopes, consistent with an increased range of grounding-line migration on flatter slopes. In Figure 3e we show the results for till extrusion but with the till being sourced only from compression at the grounding line (Figure 2d; Supplementary Video S3). As the fundamental mechanism remains

the same, the pattern of ridges is similar but overall ridge heights are slightly smaller and ridge spacing is slightly greater (Figures 3e, f). This is because the volume of sediment available only depends on the tidal amplitude and associated change

in grounding-zone width; in our experiments, the degree of compression is not varied with tidal amplitude (Sayag and Worster, 2013). Outside of areas of composite ridges, ridge spacing more obviously increases as ridge amplitude decreases (compare Figure 3e to Figure 3c) leading to more variability in correlation of ridge height and spacing when compared with the previous till extrusion mechanism ($R^2 = 0.447$; Figure 3f).

The final mechanism, resuspension in tidal cavities (Section 2.1.3), also produces one ridge at the low-tide position but with less regular shapes than the till extrusion models, notably when there are secondary peaks in the tidal amplitude and during smaller (neap) tides, which tend to occur together (Figure 3g; Supplementary Video S4). Composite forms are also modelled becoming very pronounced on lower bedslopes and forming on the transitions from neaps to springs to produce large ridges up to 1 m in height (as measured from the sea floor elevation of neap ridges). Yet ridge spacing reaches its maximum just after this leading to a poorer correlation between these metrics than for the previous mechanism (Figure 3f). Outside of composite forms, taller ridges are generally produced by larger tides (springs, Figure 3g) and on lower bedslopes because a greater area of the grounding zone is exposed and, during larger tides, the drainage rate (velocity) of water out of the cavity is higher. Together, these factors lead to a greater volume of sediment being eroded and available for ridge formation. However, superimposition of ridges formed at lower (neap) tides and at the "balance spot" described previously leads to some anomalously tall ridges spaced closely together and a poorer correlation of heights and spacing results ($R^2 = 0.239$; Figure 3h).

### 3.2 Implications for the formation of corrugation ridges at grounding lines

To assess whether the formation mechanisms modelled here are robust, we compare the model results with real-world observations from the corrugation ridges observed at Thwaites (see Section 2.2, Tables 2, S1). We directly compare modelled output with observations for 14-day periodicity, ridge amplitude, ridge height-spacing correlation, and the production of individual or composite ridge forms. We also consider, in a more qualitative way, ridge (a)symmetry and consistency of ridge morphology. We then utilise existing knowledge of glacial sediments from former grounding-line settings (e.g., Lindén & Möller, 2005; Demet et al., 2019; Smith et al., 2019) and from well-known sedimentary processes, to make inferences about the potential sedimentological properties of corrugation ridges formed by each mechanism. To date, corrugation ridges have not been directly sampled. Our observations from Thwaites and sedimentological inferences for each mechanism are presented in simplified form in Table 2, and further detail and discussion of the evidence for each mechanism supplemented by referenced background information is given in Supplementary Table S1. We consider the plausibility of each mechanism in turn.

### 3.2.1 Constant till flux mechanism

The till-flux mechanism produces mostly double-peaked or highly composite ridge forms, with no clear fortnightly pattern to either the spacing or ridge height (Figures 3a b). This strongly contrasts our Thwaites observational dataset (Tables 2, S1) with its one ridge per tidal cycle, as well as numerous other tidally modulated ridge landforms (Jakobsson et al., 2011; Graham et al., 2013; Dowdeswell et al., 2020; Supp. Text 1, Figure S1). So, while this is the simplest conceptual model, we clearly must discount it.

### 3.2.2 Till extrusion mechanism

The till extrusion models, with either a constant till flux or grounding-line compression, produce ridges that are arguably most similar to the very regular features observed at Thwaites (Figures 3c, 3e, 3i), although some differences do exist. The observed 14-ridge periodicity (Section 2.2) is reproduced, as is the normal correlation between ridge height and spacing (Tables 2, S1). As stated in Section 2.1.2, we cannot assess the symmetry of ridges produced by this mechanism because we assume that each ridge forms with the same geometry: a landward-facing side reflects the ice-shelf base cavity angle (equal to $\theta_{eff}$) and the

seaward-facing side is vertical. This is shown clearly in the model results in Figures 3c and 3e in comparison with the symmetric forms in Figure 3i. More complex modelling of the extrusion process, viscous slumping of the ridge, and ice-till coupling during the push process for a subglacial till-type of material would be required to allow a detailed comparison of ridge shape. However, we do note that when effective slopes most similar to the observed bed at Thwaites ($\theta_{eff} = 0.02$) are used, pronounced composite ridges are formed resulting in a long wavelength ($\lambda \sim 90$ m) cyclicity in bed topography (Figures

3c, 3e). By way of comparison with the Thwaites observations, although some corrugation ridges do appear to climb and descend longer wavelength topography we do not see any clear evidence for either composite ridges nor regular bed undulations in the sea floor data.

An extrusion-type model is the mechanism favoured by marine geoscientists (Dowdeswell et al., 2020; Graham et al., 2022) based on form analogy with other landforms produced at glacier and ice-sheet grounding lines. Recessional or De Geer

moraines that form by push as a grounding line readvances over a bed of unconsolidated sediments (Boulton et al., 1986; Lindén & Möller, 2005) can also have very regularly spaced, repeating morphologies when produced annually by winter readvances of local glacier fronts (e.g., Ottesen & Dowdeswell, 2006; Todd et al., 2007; Burton et al., 2016). However, these moraines tend to be at least an order of magnitude larger in their dimensions (height, width, slope angle), and are often asymmetric in cross section due to the forward motion of grounding-line readvance steepening the ice-distal ridge face

(Boulton et al., 1986). Here, the model only simulates the resettling of ice on to deformable (recently subglacial) sediments at the low-tide position squeezing sediment up into a small ridge, leaving an imprint of the grounding line on the seafloor. As such, we do not expect to recreate ridge asymmetry with this model, but note that only the landward side of the corrugation ridge should reflect the shape of the ice base as it touches down onto the bed (Tables 2, S1). We also do not expect any significant modification of ridge shape by post-formation downslope processes (e.g., slumping) owing to the incredibly shallow

slopes of their flanks: Graham et al. (2022) report median values of 0.05-0.2° for both flanks.

Supporting observational evidence for a till extrusion process of corrugation ridge formation is found in the remarkable consistency from one ridge to the next, i.e., as retreat progressed, at both Thwaites and in the Larsen Inlet (Dowdeswell et al., 2020; Graham et al., 2022). This ridge-to-ridge consistency is the case even when along-ridge form is quite variable indicating that the small-scale (metres to decimetres) shape of the ice base did not vary as the grounding line retreated (Batchelor et al.,

2020), which perhaps supports widespread dynamic thinning as a retreat mechanism over high rates of melting that might alter the shape of the ice base variably along the grounding line. A process analogy to the squeezing out of sediment that we describe

here may come from crevasse-squeeze ridge landforms, which are metres high, sharp-crested and formed of till, and are mostly observed in association with surge-type glaciers. For surging glaciers, the process is squeezing of subglacial till into basal crevasses that opened when high basal water pressures facilitate bottom-up crevassing (Rea and Evans, 2011); however, subglacial till extrusion into existing ice-margin crevasses have also been described for non-surging but radially-spreading glaciers (Evans et al., 2016).

In terms of inferred sedimentological properties of the ridges, it is well known from both extant (sampled) glacier beds (e.g., Englehardt et al., 1990; Tulacyk et al., 1998; Christ et al., 2021) and formerly glaciated terrains (e.g., Boulton et al., 1976; Evans et al., 2006; Demet et al., 2019) that subglacial sediments (or tills) typically have diamictic grain-size distributions. That is, most or all grain-size fractions are represented in a poorly sorted or homogeneous mixture often lacking specific structures or textures (e.g., Eyles et al., 1983; Clarke, 1987). With the till extrusion mechanism, either with constant till flux or grounding-line compression, we expect that subglacial till was originally supplied to the grounding zone via a deforming "conveyor belt" at the ice-bed interface (Alley et al., 1987; Kamb, 2001). If the till used in ridge formation is sourced from a constant till flux to the grounding line during its retreat then this till layer is "fresh"; conversely, if the till is sourced by grounding line compression of material over a former ice-sheet bed then the till may pre-date the grounding-line retreat phase. In either case, we do not expect any textural or grain-size differences between the ridge and inter-ridge areas beyond the deformational sediment textures and structures typically associated with subglacial traction tills (e.g., van der Meer et al., 2003; Evans et al., 2006; Reinardy et al., 2011; Table S1). Similarly, we would not expect any shearing deformation or stacking of till layers associated with forward motion, or push, as might be found within recessional push moraines or thrusts (e.g., Menzies, 2000; Evans & Hiemstra, 2007; Table S1).

Based on our model results we cannot properly distinguish between the constant till flux or grounding-line compression source for the ridge material. Both mechanisms produce a series of regular corrugation ridges, although the correlation of ridge height to spacing is stronger, and in line with our observations, for the constant till flux mechanism. We also tend to favour this mechanism for till supply because when visualising a grounding line that retreats only 6 m per day we recognise that almost the same area of sea floor will be compressed every day as the ice lifts and resettles with the tides. We posit that repeated compression of till across the grounding zone would make this material more difficult to mobilise over time (by compressing, dewatering and stiffening). Basal properties derived from seismic datasets and inferred from numerical models provide some support for this notion. At Whillans Ice Stream, arguably the best studied extant Antarctic grounding zone, seismic properties (density, $\rho$; p-wave velocity, $V_p$; s-wave velocity, $V_s$) indicate significant stiffening of a 5-m till layer across a sharp transition at the grounding zone (Horgan et al., 2021). This stiff, low permeability substrate at the grounding zone is consistent with models of ice-shelf flexure (for fixed and tidally migrating grounding lines) that predict compression, and potentially dewatering of subglacial till, immediately upstream of the grounding zone as ice bends down into the substrate at high tide (Walker et al., 2013; Sayag and Worster, 2013). Whether this compression and dewatering is enough to prevent material squeezing out from the grounding line after the ice resettles on the seafloor is unknown.

Given the above discussion, we prefer the idea of a constant till flux supplied by the "till conveyor" to form the ridges. With this in mind, we can compare previous estimates of the subglacial sediment flux (to the grounding line) with the volume of a ridge formed once per day at Thwaites Glacier to see if this mechanism of till supply is realistic. Estimated fluxes of 100 $m^3$ $yr^{-1}$ (per metre ice stream width) (0.27 $m^3$ $day^{-1}$) for the extant grounding line of Whillans Ice Stream (Engelhardt & Kamb, 1997; Anandakrishnan et al., 2007), and 800-1000 $m^3$ $yr^{-1}$ (2.2-2.7 $m^3$ $day^{-1}$) for large, fast flowing Antarctic and Greenlandic

palaeo-ice streams (Dowdeswell et al., 2004; Hogan et al., 2012, 2020) suggest that daily fluxes of sediment to the grounding line are likely to be on the order of a few tenths of one $m^3$ to a few $m^3$. If we consider the average dimensions of a triangular, symmetric corrugation ridge at Thwaites (0.2 m tall, 5 m wide at the base) then we can calculate that every 1-m section along the ridge will contain 0.5 $m^3$ of sediment. Thus, it appears that the subglacial till conveyor is able to supply about the right amount of sediment to build a daily ridge of the same magnitude as the Thwaites corrugation ridges.

One potential contradiction for the till extrusion mechanism comes from the modelling results of Warburton et al. (2020) who found that low subglacial till permeability filters the grounding-line response to tidal forcing, essentially fixing the grounding line at the high-tide position. As pointed out by Graham et al. (2022), the strong tidal periodicity of the corrugation ridge dimensions should, therefore, indicate a high permeability substrate in the region of ridge formation despite observational evidence that most subglacial tills are cohesive and likely to have low permeabilities. Graham et al. addressed this problem by

suggesting that water may drain out of the grounding zone via a series of shallow canals, although another alternative is that the pattern of grounding-line migration at Thwaites is not controlled by fluid connectivity through the till, as has been suggested for the Whillans grounding zone (Horgan et al., 2021).

### 3.2.3 Resuspension in tidal cavities mechanism

Like the till extrusion mechanism (with either till source) and with real-world corrugation ridges, the resuspension of sediment

in tidal cavities mechanism (shortened to "resuspension mechanism" hereafter) produces a series of corrugation ridges with a 13-15 cycle periodicity in their amplitudes and spacings (Figures 3g, 3h), although these are only weakly correlated (Tables 2, S1). The shape of the modelled ridges, which are sharp-crested, is also different from the observation of symmetric ridge cross-sectional shape at Thwaites (Figure 3j). Again, this is largely due to the simplicity of our model that preserves a vertical seaward slope.

Perhaps of note is that the resuspension mechanism is the only mechanism that would result in a variation in acoustic backscatter over the ridges due to a grain-size change (Tables 2, S1). Smaller particles would be preferentially eroded from the grounding-zone bed and eventually deposited to form the ridges leading to finer grain sizes in the ridges versus inter-ridge areas. However, the backscatter response on the side-scan sonar images clearly shows higher reflectivity (i.e., brighter returns) on the ridges and lower reflectivity (darker returns) in the inter-ridge areas (Figure 1b). This is the opposite of what we would

expect if the ridges were composed of fine-grained material and the inter-ridge areas consisted of the remaining coarse particles (e.g., Lurton and Lamarche, 2015). In contrast, we can envision squeezing-out of till into a ridge at the grounding line (as in the till extrusion mechanism) may produce a rougher surface texture producing relatively higher backscatter returns than the

inter-ridge areas that have been compressed, or that the stronger returns on the ridges are purely a product of the topography of the ridge fronts. A detailed sampling campaign would be required to ground truth the sedimentology of ridge and inter-ridge areas and so to determine the cause of the backscatter variations.

A sediment suspension mechanism is probably the easiest to interrogate using a combination of empirical datasets and established sediment dynamics theory. Our model simulates water rushing in to (and out of) a 1-km wide grounding-zone cavity when set up in a Thwaites configuration and assumes that enough particles can be eroded from the bed of the grounding zone to build a ridge every day. Precedence for such a mechanism comes from the process of "tidal pumping", whereby tidal flows in a sub-ice-shelf cavity winnow fine-grained particles from glacigenic debris as it melts out from the ice-shelf base near the grounding line and then transports these particles in suspension seaward, eventually depositing them as laminated clayey to fine sandy units. This tidal pumping is well established based on marine sediment core data from the Antarctic continental shelf (Domack & Harris, 1998; Domack et al., 1999) and provides some support for the notion that tidal water velocities can be at least high enough to transport grains up to fine sands in size (0.25 mm diameter). Thus, our mechanism here is similar to tidal pumping because we stipulate (in the model) that the outflows carrying fine-grained particles do not migrate vertically, exiting the grounding zone horizontally and immediately depositing their suspension load to form a ridge.

Accepting this and remembering that every 1-m section along a corrugation ridge will contain around 0.5 m$^3$ of sediment, a basic calculation can be done to determine how much subglacial sediment would need to be eroded from the bed to produce a 0.5 m$^3$ ridge after each tidal cycle (i.e., once per day at Thwaites). Subglacial tills recovered from the western Amundsen Sea in marine sediment cores have similar grain-size distributions with around 90% of grains (by weight) being smaller than 2 mm in diameter, and around 65% smaller than 0.25 mm (Smith et al., 2011; see Supplementary Table S2). Using a wet bulk density of 1.575 g cm$^{-3}$ for a standard marine sandy-silty clay (Hamilton and Bachman, 1982), we calculate that around 788 kg of material is required for each 1-m section of ridge. Assuming that grains are eroded across the entire grounding zone, and that only 65% of them are small enough to be transported to form a ridge, then we calculate that grains would only need to be eroded from the upper ~0.6 mm of the till (wet bulk density 2.1 g cm$^{-3}$; see Supplementary Text B) to provide enough material to form each ridge. This thickness is very small indeed and may indicate that there would have been a plentiful supply of fine-grained particles (from the bed) across the grounding zone with which to form the ridges. However, there are several potential pitfalls with this simple calculation: (1) with a grounding-zone width of 1 km tidal water velocities are extremely low, on the order of only a few cm s$^{-1}$, and therefore below the speeds required for the erosion and transport of fine sands from the bed (e.g., McCave & Hall, 2006); (2) it is unlikely that particles would be resuspended from across the entire 1-km wide grounding zone because as the ice-shelf base becomes less confined (away from the grounding line) the influence of tidal flows would diminish; and (3) it is unclear how fine-grained particles would continue to be mobilised from sub-seafloor depths once surface sediments had been winnowed presumably to form either a hardground or lag deposit as is found when strong ocean bottom currents winnow seafloor sediments (Anderson et al., 1980; Hillenbrand et al., 2003) or to create a granulated facies (with fines removed) similar to that found overlying subglacial tills elsewhere in Antarctica (Domack & Harris, 1998 ; Domack et al., 1999; Kirshner et al., 2012).

The first of these issues is probably the hardest to counter. Water velocities of only a few cm s$^{-1}$, which are in fact in line with the few existing hydrographic observations of extremely low to undetectable tidal current velocities in grounding-zone cavities (Begeman et al., 2020; Davis et al., 2023), would only allow for transport of the very finest particles (clay and fine silts <10 μm). Velocities above ~20 cm s$^{-1}$ are required to keep sand-sized grains (diameters 63 μm - 2 mm) mobile in suspension (McCave & Hall, 2006). Such velocities could be achieved if the grounding zone was much wider; for example, a 10-km wide grounding zone, which is at the upper bound of observed widths around Antarctica (see Brunt et al., 2011), could produce tidal water velocities of ~23 cm s$^{-1}$ and, therefore, transport sand-sized grains. The occurrence of granulated units elsewhere on the Antarctic continental shelf, including in the Amundsen Sea (Kirshner et al., 2012), might actually indicate that such velocities can be achieved. Turning to the second and third issues, if grains were only eroded from a 100-m wide area adjacent to the grounding line then all particles from the upper ~6 mm of the bed would be required to build a ridge and mobilising grains from sub-seafloor layers becomes increasingly difficult to envision (recalling daily retreat rates of only 6 m so the bed would be repeatedly eroded). If grains can only be mobilised from the newly exposed grounding zone area (6 m wide) then the depth of erosion increases to an untenable 625 mm (62.5 cm). Even if we accepted this erosion process for some area of the grounding zone, it is also difficult to see how such a "dumping" mechanism would result in all grains being deposited instantaneously (the finest particles would surely be carried further seaward), or how this would produce such consistent ridge-to-ridge morphologies as is observed at both the Thwaites and Larsen Inlet sites (Table S1).

Thus far, we have only considered the transport of grains in suspension and not how easy it is to erode them from the bed. The matrix of subglacial tills, including those recovered from the Amundsen Sea area close to Thwaites Glacier (Smith et al., 2011; 2013; Table S2) and those observed directly at the few Antarctic grounding zones to have been accessed (Langovde Glacier, East Antarctica: Sugiyama et al., 2014; Mackay Glacier: Powell et al., 1996), are dominated by fine grain sizes (typically ~55-70% clays and silts). As a result, subglacial tills behave cohesively (with respect to erosion by water), and the fine grains within them are able to withstand larger values of shear stress without being eroded (Mier & Garcia, 2011). Unidirectional flume experiments designed specifically to assess erosion of glacial till or till-like sediments in rivers or coastal environments have returned critical shear stress values of 4-9 N m$^{-2}$ to initiate erosion (McNeil et al., 1996; Mier & Garcia, 2011; Pike et al., 2018), whereas our model predicts a shear stress across the grounding-zone bed of only ~1 N m$^{-2}$. Although this may intuitively indicate that greater shear stresses (or water velocities) are required to erode glacial till across the grounding zone the additional complexities of being in a grounding-zone setting (tidal currents switching direction and lift-off and re-settling of the ice sheet compressing but also disturbing the bed every 12 hours for diurnal tides as in the Thwaites area) mean that this assumption is not straightforward. In addition, we note that there is a growing body of observations showing that some Antarctic grounding zones are in fact kilometres wide (Mohajerani et al., 2021; Milillo et al., 2022; Chen et al., 2023), an order of magnitude greater than predicted for ice shelves in hydrostatic equilibrium (Mohajerani et al., 2021). Such a wide grounding zone would significantly increase tidal flow velocities, especially in narrow cavities, and potentially allow for much greater erosion and sediment transport. Furthermore, current erosion of the bed may not be the only source of sediment to the tidal cavity. Meltout of basal debris from the ice-sheet base as the cavity opens could be an additional source of particles (see the tidal pumping

mechanism described above) although this may be limited to areas very close to the grounding line; Domack and Harris, 1998; Smith et al., 2019), and the disturbance or ploughing of the bed by a rough ice base swiftly followed by lift-off as the tidal cavity opens is another process that may mobilise fine-grained sediments into suspension, therefore lowering the current velocities required to transport material out of the grounding zone.

The tidal resuspension mechanism Is eminently testable with direct sampling. Sediment samples from an area of corrugation ridges should comprise normally sorted, fine-grained deposits in the ridges, and winnowed subglacial sediments (with fines removed at least in their uppermost layers) in intervening bed areas (Supplementary Table S1). Although it appears that there is sufficient supply of fine-grained sediments to the grounding zone for this mechanism to be plausible, the cohesive nature of glacial tills and low predicted tidal current velocities makes it difficult to see how enough material could be mobilised from

the bed every day to produce even the subtle (low amplitude) ridges described from Thwaites or the Larsen Inlet.

### 3.2.4 Implications for bed geometry

A final yet important consideration is how the models perform with realistic ice-bed geometries, which is represented in the models by $\theta_{eff}$. Both the till extrusion and resuspension mechanisms produce tidally correlated ridges for the larger values of $\theta_{eff}$, but increasingly large composite ridges with a decreasing number of individual ridges, quite unlike the observations that

consistently show individual ridge forms were produced at each tidal cycle on low-gradient bedslopes (i.e., smaller values of effective slope; see Section 2.2). This conflicts with the suggestion in Graham et al. (2022) that the corrugation ridges at Thwaites may have formed in ice-plain conditions ($\theta_{eff}$ close to 0) because the thinning or melting rate required to sustain rapid retreat is proportional to $\theta_{eff}$ and is thus easier to achieve for lower values of effective slope. However, the control on composite ridge formation is the balance between the daily retreat rate (which we take to be directly observable), and the changes in

grounding-line migration range over the tidal cycle, which depends only on the tides and the effective slope. While it is possible that present tides do not represent the tidal amplitudes during the time of ridge formation, we assume that $\theta_{eff}$ is left as our only control parameter. We therefore suggest that it would be impossible to form the observed ridges at small $\theta_{eff}$, implying that a large thinning rate was truly necessary to drive the retreat. As such we reiterate that, in addition to the melting at the grounding line quantified by Graham et al. (2022), dynamic thinning following ice-sheet acceleration likely also contributed to the total

thinning rate.

### 3.3 Larsen Inlet corrugation ridges: model validation

A natural question for model validation is whether the mechanisms described above can produce corrugation ridges in other settings, if the models are run using parameters other than those representative of conditions at Thwaites Glacier. The Larsen Inlet corrugation ridges (Dowdeswell et al., 2020) represent distinctly similar landforms in a somewhat different setting – the

475 ridges are on a shallow landward slope (typical slopes of 0.5-1° which equates to lower values of $\theta_{eff}$), separated by 20-25 m, in a location where there are two tides per day (corresponding to a suggested retreat rate approximately 10 times larger than at

Thwaites). In addition, some of the ridges are double-peaked (M-shaped), reminiscent of the output with the constant till flux mechanism at Thwaites (e.g., Figure 3a).

In Figure 4a we show the results of the three models run using Weddell Sea tides (for the Larsen Inlet site), a retreat rate of 50 m per day (as suggested in Dowdeswell et al., 2020), and $\theta_{eff} = 0.02$. The most notable result is that despite two tides per day, over much of the tidal cycle only one ridge is preserved per day, leading to a ridge spacing that is double the observed values. This is due to the difference in amplitude between the two daily tides, so that alternating low-tide positions eliminate the previous, less far seaward ridge. Figure 4b shows the result of the same modelling but with a retreat rate of 25 m per day, which is still extremely high compared to all but the fastest modern retreat rates (c.f. Milillo et al., 2022), but half the value previously associated with these bedforms. With this reduced retreat rate, we do reproduce the correct mean corrugation ridge spacing. Further, we see that the constant till flux mechanism still does not produce coherent tidally modulated ridges, while both the till extrusion and sediment resuspension mechanisms produce clear corrugation ridges. We also show that both these preferred mechanisms produce occasional but discrete double-peaked ridges (see arrows in Figure 4b), as are often observed at Larsen Inlet (Figure 4c; Table S1), but which were not formed by the models under Thwaites parameters. This reinforces our view that either till extrusion or sediment resuspension represent the most plausible way to form corrugation ridges, whilst also building confidence that our modelling is not overly tuned to Thwaites-like conditions and so may apply more generically around Antarctica.

### 3.4 Observations necessary to further refine formational mechanisms

In considering all the above discussed mechanisms, we suggest that future work into corrugation ridge landforms, their geometries, and composition should be undertaken to explore, test and develop the ideas presented here. One observational approach might be to assess contemporary relationships between grounding-line behaviour and tides, in an active ice-stream grounding-zone setting. A candidate environment would be a part of the modern grounding zone where the glacier is presently sat on a flat bed and is measured to be retreating rapidly over short timescales (e.g., Pope Glacier, one of Thwaites' neighbouring glacial outlets, which retreated at rates of >10 km yr$^{-1}$ in 2017 (equates to ~27 m day$^{-1}$); Milillo et al., 2022). However, these surveys remain a significant challenge given the extreme difficulty in accessing grounding zones around the Antarctic, either from the ocean or from the ice-sheet surface above. Two alternatives we propose are: (1) to recover further high-resolution geomorphological information and observations at ice-proximal sites that represent very recently deglaciated proglacial seascapes (c.f. Graham et al., 2022); and (2) to seek out candidate grounding-zone wedges on the continental shelf in the open ocean, that might preserve additional evidence for rapid retreat phases in their seafloor landforms (c.f. Dowdeswell et al. 2020; Batchelor et al., 2023). In both instances, because of the fine-scale of the landforms in question, we argue that even mid-water AUV geophysical mapping is still insufficient to fully understand the mechanisms in question. Remotely operated vehicles (ROVs) with camera capabilities, as well as low-altitude bathymetric LIDAR scanning equipment, would provide the optimum platform for future investigation. Furthermore, in both cases, we recognise that many of the key tests of corrugation ridge formation rely upon physical sampling of the features. Equipping seafloor exploration vehicles with mini-vibrocoring

tools would allow for the recovery of sediments from targeted locations across individual corrugation ridge landforms that can test some of the hypotheses put forward in this paper.

## 3.5 Potential for further modelling work

The work presented in this paper is a first quantitative modelling effort for several previously conceptual models of corrugation ridge formation (at grounding lines) and has improved our understanding of which mechanisms are most plausible, and which
parameters (surface slope, average till flux, mean retreat rate) control the basic components of ridge morphology, such as amplitude and spacing. However, these models rely on several simplifying assumptions, and do not attempt to replicate the dynamics of ridge formation. As a result, we cannot capture the detailed shape of the ridges, a significant aspect of the observational data that we are presently unable to compare to. Having identified the till extrusion and resuspension models as most able to produce something akin to the real-world tidally modulated corrugation ridges, future modelling should focus on
the dynamics of these mechanisms.

In the resuspension mechanism, the parametrisation of sediment erosion could be improved by incorporating more detailed modelling of tidally-driven flows between the ocean and the grounding-zone cavity, rather than assuming a sudden transition in the strength of the flow. With a more detailed understanding of the flow speeds throughout the grounding zone, a future model could use the implied spatially and temporally evolving grain-size dependent sediment flux to model the shape of
depositional ridges and to quantify the expected grain-size sorting.

For the till extrusion mechanism, we have neglected the details of the flow field in the till. Future modelling should include the viscous and elastic coupling between the ice and the till, which controls both the till flux towards the grounding line and the shape of the base of the ice, where we have taken these as given, and as constant. This modelling could also capture ocean-till interactions at the grounding line as the ice retreats, and the viscous relaxation of the ridges after their formation. These
processes all depend sensitively on the till rheology and matching the final shape of the ridges to the observations would therefore help to constrain parameters such as till yield strength and compressibility.

An additional target for future modelling work could be to simulate corrugation ridge formation within iceberg ploughmarks as individual icebergs periodically impact the seafloor (e.g., Figure S1d). Although the details of this process are different from our grounding line setting, i.e., there is no sediment supply from upstream nor compression from a lifting and settling ice sheet,
the relative simplicity of this process may help to support a till extrusion (squeeze out) mechanism at grounding lines.

## 4 Conclusions

Using mathematical modelling we have investigated three formation mechanisms (constant till flux, till extrusion, and resuspension in tidal cavities) for low amplitude, regular corrugation ridge landforms at Antarctic grounding lines. The models produce plausible ridges for the only two real-world examples of grounding-line corrugation ridges currently known, at
Thwaites Glacier, West Antarctica and in the Larsen Inlet, eastern Antarctic Peninsula, using different boundary conditions

(bedslopes, tides), and indicate that the till extrusion (squeeze out at the grounding line as the ice re-settles on the seafloor at low tides) and resuspension (erosion and flushing out of fine-grained sediments by tidal flows) mechanisms produce ridges most similar to observed landforms. After comparisons with empirical datasets from extant and relict grounding zones, previous modelling studies, and known sedimentary processes, outstanding questions remain around the fine-grained sediment

supply (erodibility of tills) for the resuspension mechanism, as well as how it would produce such consistent ridge morphologies. As such, the till extrusion mechanism is the preferred mechanism for corrugation ridge formation.

Given the significance of these landforms as specific markers of quantifiable rapid grounding line retreat behaviour, probably with a strong contribution from dynamic thinning, we advocate for their further exploration. High-resolution seafloor surveying by AUVs or ROVs, and precision seafloor sampling on ridge crests and inter-ridge areas on a transect across the ridges–

followed by detailed sedimentology and dating–- would confirm the tidal modulation of the ridges and identify the specific times (and climatic conditions) when rapid grounding-line retreat occurred, as well as help determine the formation mechanism for these intriguing landforms. Additional mathematical modelling, guided by such observations, testing different sediment properties (e.g., permeability, grain-size distributions, cohesion), variable boundary conditions (e.g., ice-shelf cavities and bed shapes), and an elastic (bending) ice shelf should help to determine which grounding-zone conditions and geometries were

present during the formation of these landforms, and therefore which grounding lines may be vulnerable to (or allow) the very rapid grounding-line retreat (kms per year) associated with them.

### Code and Data availability

All data needed to evaluate the conclusions in this paper are presented in the paper or in the Supplementary Information including the model code. The model code is also available at the University of Cambridge Apollo Repository (DOI

forthcoming). A high-resolution multibeam raster grid over the Thwaites Glacier corrugation ridges (AUV mission 009) is available for download from the Figshare repository (https://doi.org/10.6084/m9.figshare.20359920.v1); for full details of this dataset see Graham et al. (2022). Gridded multibeam data for the Larsen Inlet corrugation ridges is available upon request from JAD.

### Author contribution

KAH, KW, AG and RDL designed the study; KW wrote the model code and performed the simulations with contributions from JN and DH; JAD provided the data from the Larsen Inlet; KAH and KW wrote the manuscript draft with contributions from AG; RDL, JN, DH, and JAD reviewed and edited the manuscript.

**Competing interests**

The authors declare that they have no conflict of interest.

**Acknowledgements**

This work was supported was supported by grants NE/S006664/1 (KAH, RDL), NE/L002507/1 (KW), and NE/S006206/1 (AG) from the Natural Environment Research Council. Seafloor data collection in the Larsen Inlet, western Weddell Sea was funded by the Flotilla Foundation. The authors thank the masters, officers and crews of the R/V *Nathaniel B. Palmer* and *S.A. Agulhas II*, as well as the science parties of cruises NBP19-02 and The Weddell Sea Expedition 2019 for their valuable support
during AUV data collection. We thank Anna Wåhlin, University of Gothenburg, for her generous contribution in collecting the downward looking AUV data at Thwaites Glacier during NBP19-02 and for her comments on this manuscript; Claus-Dieter Hillenbrand and James Smith, British Antarctic Survey, are thanked for helpful discussions relating to the sedimentology of Antarctic tills and glacial sedimentary processes; Ed Self, Gardline Ltd., is thanked for access to the bathymetry dataset in Supplementary Figure S1d. This work is also an output of the Thwaites Offshore Research (THOR)
project, a component of the International Thwaites Glacier Collaboration (ITGC). Logistics for ITGC were provided by the NSF United States Antarctic Program and the NERC British Antarctic Survey. ITGC contribution no. ITGC-103.  Finally, we thank one anonymous reviewer, Sarah Greenwood and Chris R. Stokes (editor) for their helpful comments that improved the manuscript.

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

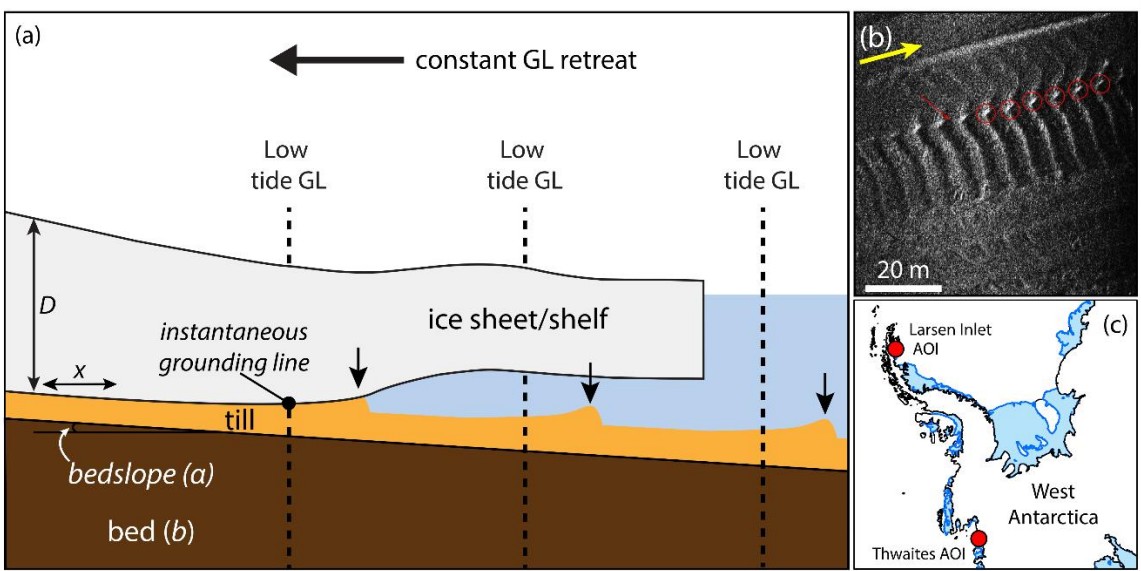

Figure 1: (a) Schematic diagram of an ice-sheet grounding line (GL) that is retreating at a constant rate but is also subject to tidal flexure that produces a low-amplitude ridge of sediment (black arrows) at each low tide grounding-line position. (b) AUV sidescan sonar image of observed corrugation ridges from a former grounding zone of Thwaites Glacier; each ridge represents a former low-tide grounding-line position. Yellow arrow shows grounding-line retreat direction; red circles highlight sediment "beads" on corrugation ridges as the ridges cross a glacial lineation. (c) Location of the Thwaites and Larsen Inlet areas of interest (AOIs) 26 km and 42 km beyond the modern glacier grounding zones, respectively.

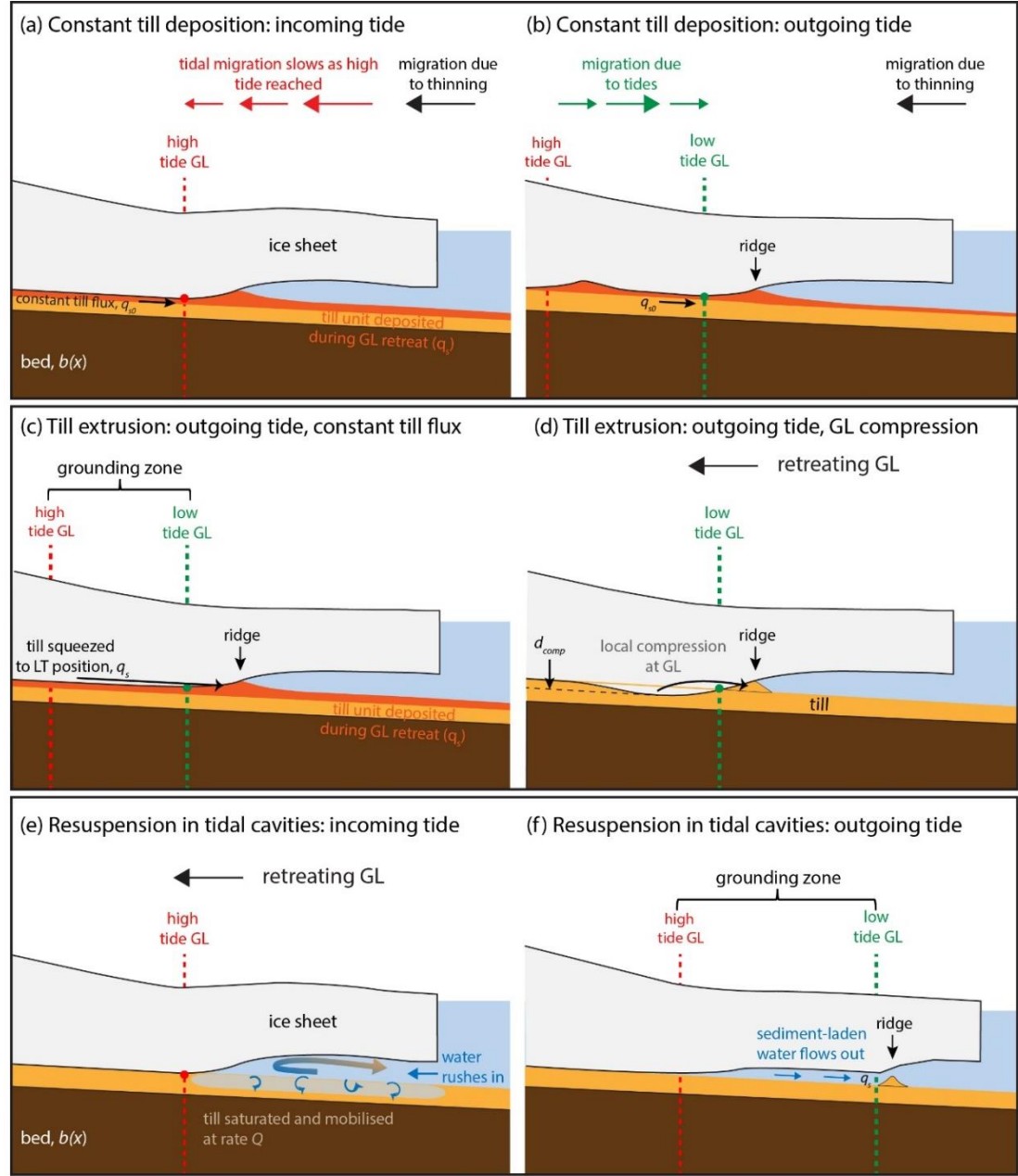

**Figure 2: Schematics of corrugation ridge formation for the mechanisms described in this paper. (a, b) Constant till flux and deposition. (c) Till extrusion with constant till flux, and (d) with grounding-line (GL) compression. (e, f) Resuspension in tidal cavities.**

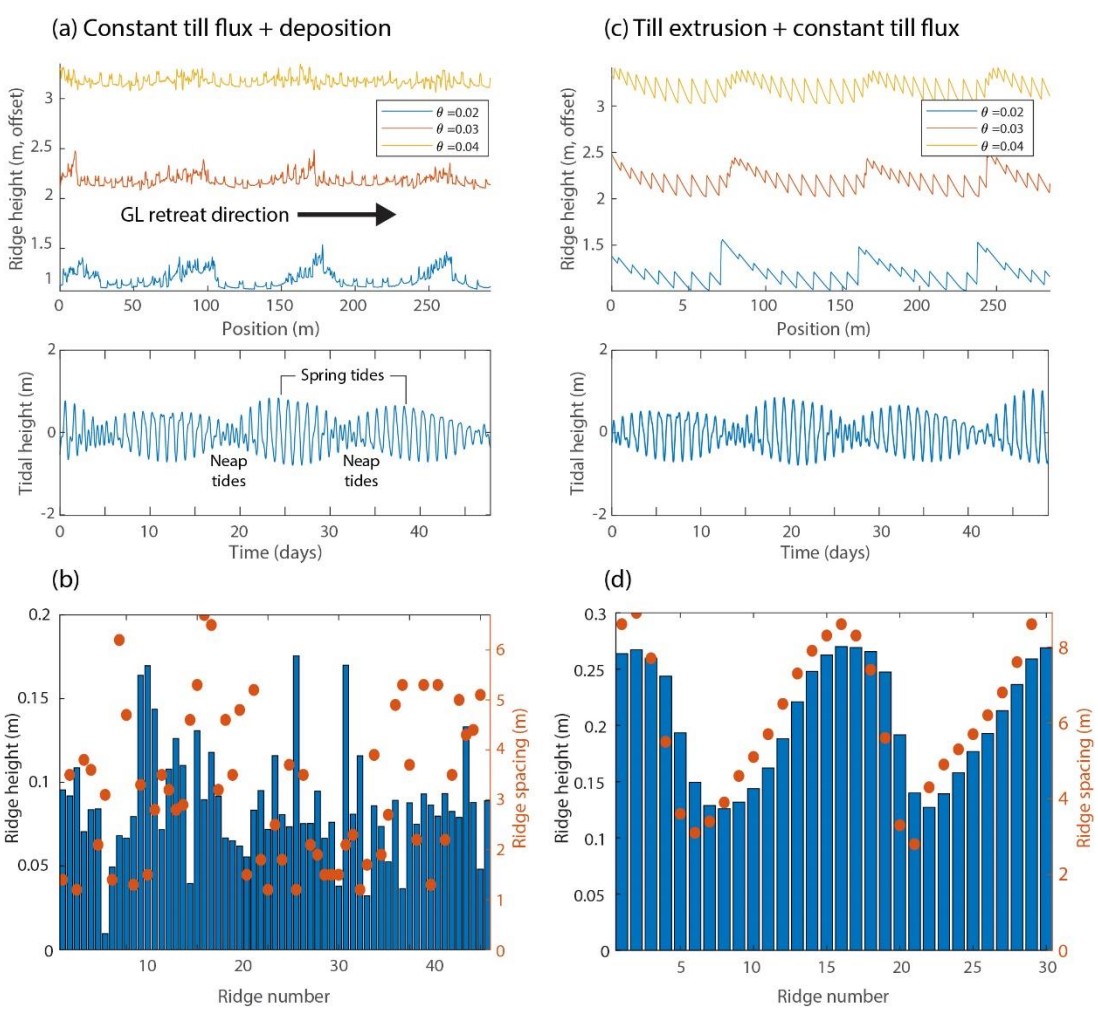


Figure 3: Results of modelled corrugation ridges formed by the three mechanisms. (a) Bed and ridge profiles (offset from each other) for different effective bedslopes using the "Constant till flux and deposition" mechanism. (b) Ridge heights and spacings for ridges with $\theta_{eff} = 0.04$ in (a). (c) Bed and ridge profiles for the "Till extrusion with constant till flux" mechanism. (d) Ridge heights and spacings for ridges with $\theta_{eff} = 0.04$ in (d). Note the different vertical scales of (b) and (d).

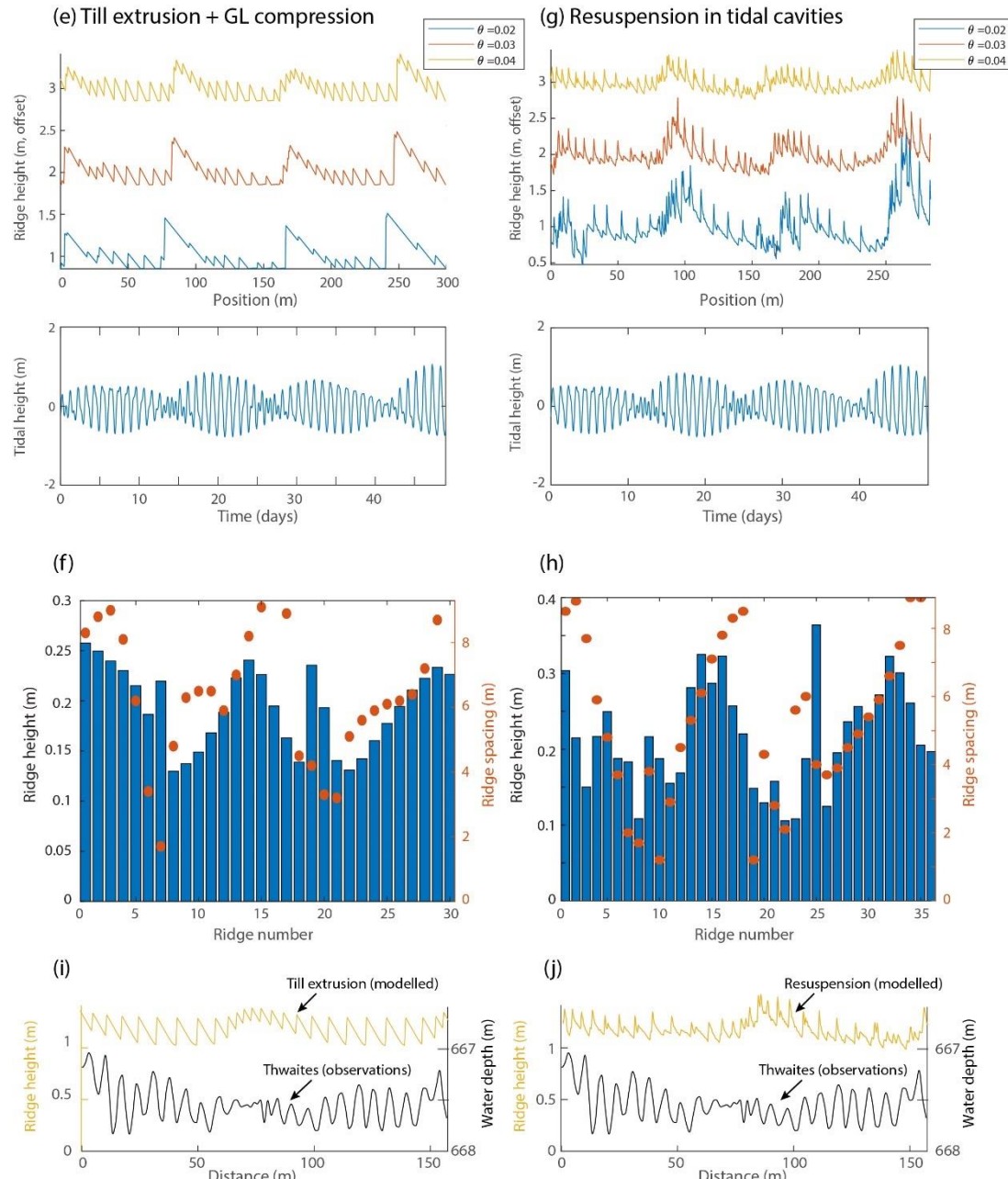

**Figure 3 cont.:** (e) Modelled bed and ridge profiles using the "Till extrusion with grounding line compression" mechanism. (f) Ridge heights and spacings for (e). (g) Bed and ridge profiles for the "Sediment resuspension in tidal cavities" mechanism. (h) Ridge heights and spacings for (g). (i) and (j) Comparisons of modelled and observed corrugation ridges: (i) till extrusion with constant till flux and a sea floor profile of ridges at Thwaites Glacier, (j) sediment resuspension in tidal cavities and the sea floor profile from Thwaites. The Thwaites profile has not been detrended and model results are shown for $\theta_{eff} = 0.04$.



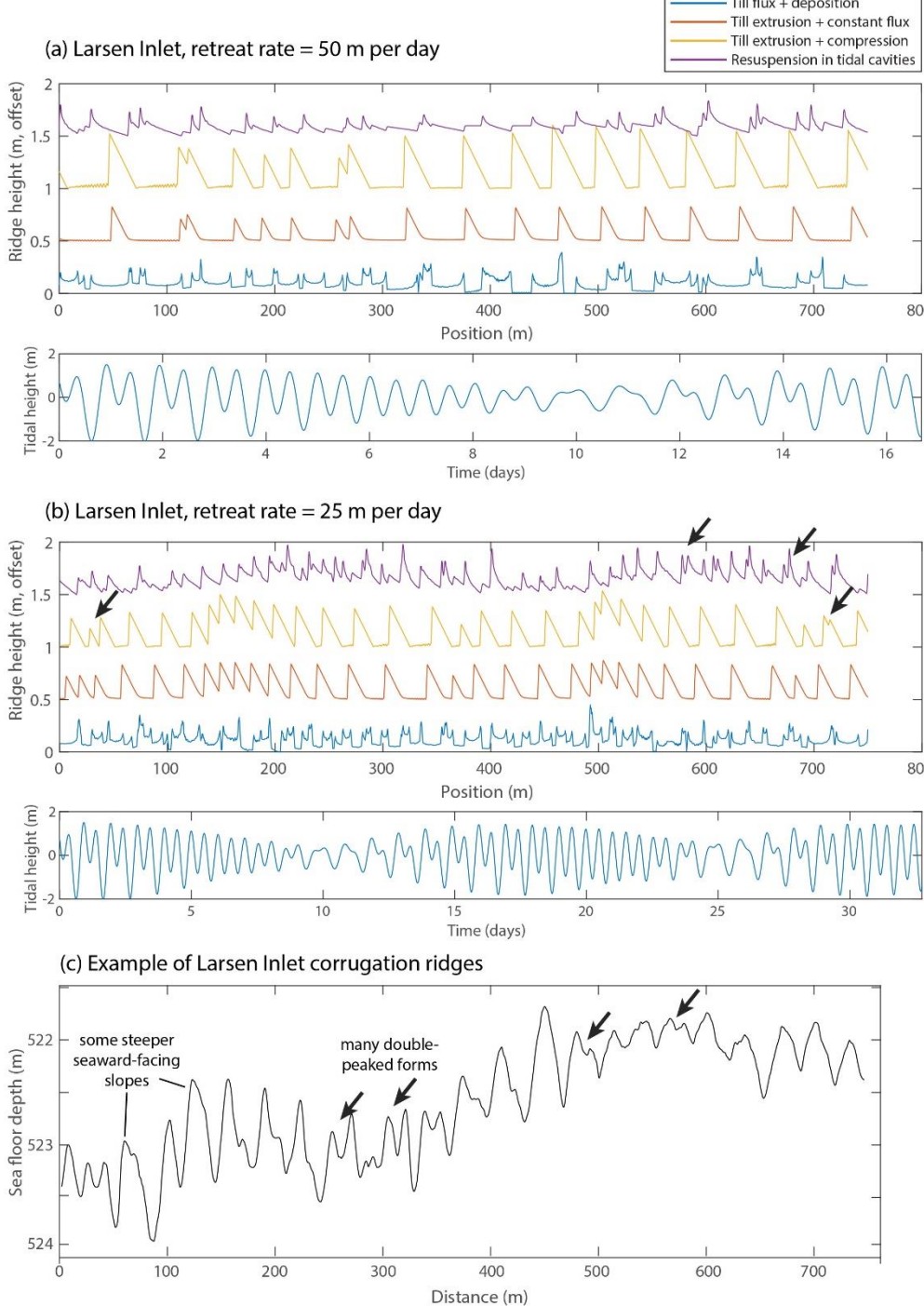

**Figure 4: Modelled corrugation ridges in the Larsen Inlet study area produced by the three mechanisms and $\theta_{eff}$ = 0.02 for: (a) a "fast" grounding-line retreat of 50 m day$^{-1}$ as suggested by Dowdeswell et al. (2020); and (b) a grounding-line retreat of 25 m day$^{-1}$. Black arrows show double-peaked or M-shaped ridges discussed in the text. (c) Sea floor profile over Larsen Inlet corrugation ridges. Note that the ridges shown have not been detrended nor lined up with modelled forms or tides for the Larsen Inlet.**

| Symbol | Parameter | Value [Units] |
|---|---|---|
| $D$ | Ice thickness | [metres] |
| $x$ | Horizontal distance (along flow) | [metres] |
| $h$ | Ocean height (changes with tides) | [metres] |
| $GL$ | Grounding-line position (migrates with tides) | [metres] |
| $r$ | Ice-sheet thinning rate (constant for ice shelf, ice sheet areas) | [m second$^{-1}$] |
| $t$ | Time | [seconds] |
| $\rho_i$ | Density of ice | 918 kg m$^{-3}$ |
| $\rho_w$ | Density of seawater | 1025 kg m$^{-3}$ |
| $\theta_{eff}$ | Effective slope | [radians] |
| $q_s$ | Total subglacial till flux to the grounding line (including effect of grounding line dynamics) | [m$^3$ second$^{-1}$] |
| $q_{so}$ | Background subglacial till flux to the grounding line (from upstream till dynamics) | [m$^3$ second$^{-1}$] |
| $\delta$ | Dirac delta function | Dimensionless |
| $d_i$ | Height of ice base | [metres] |
| $d_s$ | Height of sediment surface | [metres] |
| $d_{comp}$ | Maximum compression of ice-sheet bed at grounding line | [metres] |
| $d_\infty$ | Compression of ice-sheet bed far upstream of the grounding line | [metres] |
| $V_{rib}$ | Corrugation ridge volume | [m$^3$] |
| $\alpha$ | Angle between ice-shelf base and seafloor* | [radians] |
| $Q$ | Prescribed sediment erosion rate in ice-shelf cavity (constant) | m s$^{-1}$ |
| $T$ | Time at which the tide is highest before time t | [seconds] |
| $\tau$ | Shear stress across bed in ice-shelf cavity (changes with tides) | [N m$^{-2}$] |
| $\mu$ | Viscosity of water | [N.s m$^{-2}$] |
| $U$ | Speed of ocean height change with tides | [m s$^{-1}$] |
| $B$ | Bending stiffness (for ice sheet as an elastic beam) | [N m$^{-2}$] |
| $g$ | Acceleration due to gravity | 9.81 m s$^{-2}$ |

**Table 1: Definitions, values and units of parameters used for the modelling. *Note that for an ice shelf that is in hydrostatic equilibrium the angle α will be equal to the $\theta_{eff}$ ; this is assumed for the models presented here.**


| | Observations from Thwaites corrugation ridges or *inferred ridge properties* | Constant till flux | Till extrusion | Till extrusion with compression | Resuspension in tidal cavities |
|---|---|---|---|---|---|
| **RIDGE MORPHOLOGY** | **Frequency:** One ridge forms per day at low-tide position | ✗ | ✓ | ✓ | ✓ |
| | **Amplitude:** 13-15 cycle periodicity (assumes largest ridges form during largest tides) | ✗ | ✓ | ✓ | ✓ |
| | **Spacing:** 13-15 cycle periodicity (assumed greatest spacing during largest tides) | ✗ | ✓ | ✓ | ✓ |
| | **Correlation:** of ridge amplitudes/spacings | ✗ | strong | weak | weak |
| | **Symmetry:** ridges appear symmetric in profile | ✓ | ✗ | ✗ | ✗ |
| **OTHER** | **Acoustic backscatter:** Ridges return different BS values compared to surrounding seafloor | ✗ | ✗ | ✗ | ✓ |
| **SEDIMENTOLOGY** | *Grain sizes: Similar to subglacial till* | ✓ | ✓ | ✓ | ✗ |
| | *Sorting: Transport is by ice, no sorting* | ✓ | ✓ | ✓ | ✗ |
| | *Sedimentary structures: Deformation structures related to ice settling/push* | ✗ | ✓ | ✓ | ✗ |

**Table 2: Comparison between observed corrugation ridge properties at Thwaites Glacier and modelled ridge forms. Where ridge properties are inferred from known sedimentary processes (rather than observed directly) they are italicised.**