# Peer review of "Towards modelling of corrugation ridges at ice-sheet grounding lines"

_The Cryosphere, 2022_

## Referee Comment (RC2)

**Towards modelling of corrugation ridges at ice-sheet grounding lines**

*Hogan et al.*

This is a well-constructed manuscript presenting an elegant model–data comparison study into the formation of relatively newly-discovered, delicate grounding line landforms, which have been interpreted as forming daily due to tidal influences on the grounding zone. Verification that tidally modulated grounding line processes – and better understanding of specifically which processes – can produce discrete grounding line landforms is a significant advance that will allow us to better use the palaeo-record to reconstruct past grounding line retreat rates and dynamics and, critically, place bounds on what can be expected in terms of retreat rates from modern ice sheet grounding lines. Here, the different models and their results are well-explained, presented and evaluated; the interpretations and conclusions reached seem sound. (Note that I'm not in a position to comment on the physical formulation of the models.) The limitations of the models (conceptually in set up and in the results they produce) are thoroughly discussed, and assumptions are clearly stated, ensuring that what this work does and does not yet achieve are well-framed. I have a few rather minor comments.

Main comments

1) You state as a given that corrugation ridges are produced at grounding lines. There is good reason for adopting this interpretation for Thwaites and Larsen – occurrence on wedge surfaces, draping MSGLs, displaying lateral continuity – and therefore the approach and research framework here is justified and important. However, other examples have been reported (including by some of the authors) where corrugations are inferred to form in other ways/settings (e.g. not on wedges, and within curvilinear scours attributed to ice shelf or iceberg keels). My reading of the recent Dowdeswell & Graham papers suggests these other interpreted settings are not rejected by new observations, rather that it appears that corrugations might form in various tidally-modulated contexts for re-grounding. I think a couple of additional sentences acknowledging this (for example, around lines 45-52) would be helpful; that the Thwaites and Larsen examples likely formed at the retreating grounding line makes them especially significant and important to understand in terms of formation processes.

A follow-on from this distinction is whether the presence of corrugations in other interpreted settings (e.g. iceberg/shelf keels, forward moving) could help us (further) reject/support different mechanisms. Similarity of form and distribution might suggest that a similar mechanism, in some way associated with tidal settling, generates comparable ridge assemblages, whether by a partially grounded sheet or a re-grounding shelf or berg. But not all mechanisms may operate in different settings, or do so with rather different boundary conditions. For example, till flux isn't relevant to iceberg re-grounding cases, which might support its irrelevance, whereas extrusion (and tidal currents – resuspension?) likely does operate. Can this help your discussion?

Note that this is also relevant to the conclusion ("only two real-world examples currently known") – yes, from a grounding line, but a little mis-leading regarding the general form.

2) On first reading, I thought that the observational data record is not presented explicitly enough in the main manuscript for the reader to follow which real-world characteristics and observations are involved in the evaluation of the model output. Data-model comparison is the crux of the paper, but the relevant observational data is somewhat hidden in Table 2 and the supplementary material – there is no section in the text that explicitly states "this is what we see in the observational record". I think some minor text adjustments can improve this, both in Section 2.2 where you outline the data, and in Section 3.2 which is set up as where model output is compared to data – I've made specific suggestions below. But I also think an extra panel in Figure 3 and Figure 4 displaying a topographic profile across the Thwaites and Larsen series of corrugations, respectively, would really help. (The figure caption can make it clear what aspects of morphology it is unreasonable to expect models to replicate, in case this is a concern.)

Text suggestions:

192-198: I think the final sentence here, and thus all the relevant observational data that serves as a basis for evaluation, gets a bit lost. You could lead earlier (e.g. line 192) with a more active statement, to the effect that morphology and spacing observations from the Thwaites data (Graham et al) are given in Table 2, and serve as metrics to be compared against / to evaluate model output. 164 ridges have [… pertinent characteristics…]. You additionally consider morphological elements such as …  You could also note, here or at the opening of section 3.2, that while Supplementary Table 1 describes observed morphology as fully as possible, the main traits used for model evaluation are summarised in Table 2.

255-262: you could state in this opening paragraph to 3.2 which key observational data characteristics you will use to evaluate model output. Picking through the following sections, it seems to me these include 14-day periodicity, ridge height–spacing correlation, production of individual or composite forms … These are the main characteristics you've reported from model output, but then you also consider (a)symmetry of form, repeatability of form, ridge volume (based on cross-profile height and width)… It would help the reader to allude to these in an introductory way, so we know what to expect from the following analysis, whether just listing in-text as I have done here, or by giving a short summary of what the main characteristics of the observational data actually are.

Sections 3.2.1, 3.2.2, 3.2.3: these sections are framed as comparing model output to real-world data. 3.2.1 does just this (or begins to), as does the opening paragraph of 3.2.2; 3.2.3 doesn't make such explicit comparisons of ridge morphology or distribution at all, but rather discusses other aspects of the viability of this mechanism. These additional aspects are interesting and important, but beginning each section with a more systematic data-model evaluation would help the overall logical flow and structure of the discussion. Note that even in 3.2.1, you don't directly call upon all your observations to reject the model: state clearly, for example, that observed ridge morphology doesn't match the modelled morphology (composite forms, correlation of height with spacing…) – help the reader work through Table 2, for each modelled mechanism.

3) While Section 3.1 reports the output of the model experiments, in the final paragraph (lines 242-253) you move on to a comparison of one particular aspect (production of composite ridges and relationship with slope) with real-world data. Yet you haven't yet introduced any other explicit model-data comparison. This paragraph seems a bit out of place, since the section structure is otherwise clear and logical; the phrase "quite unlike the observations", while you haven't drawn the

reader's attention to any observations yet, highlights this section being out of place. I would consider bringing this paragraph into Section 3.2, either up front or towards the end.

4) It is not clear in your discussion of the till extrusion model (3.2.2) to what extent your discussion refers to extrusion with a flux, or to extrusion with compression. The opening paragraph (line 270-280) seems to suggest that the bed slope effect (composite ridges) should rule out extrusion with a till flux, but that is not explicit as you move through the rest of the section and discuss (a)symmetry, and consistency of form from ridge to ridge. Are you arguing in these cases that extrusion-with-flux and/or extrusion-with-compression are (or aren't) valid? (This is another example of where a more structured or systematic work-through of key traits in Table 2 vs each mechanism – see my second point, above – could help.)

From line 302 to 322, you then go on to discuss till supply as a "conveyor belt" and other reported estimates of basal sediment flux. In this discussion I no longer follow which modelled mechanism(s) you are using these "flux" estimates and concepts to evaluate: constant flux, extrusion-with-flux, extrusion-with-compression? What does "this" refer to in line 314? (In fact the whole phrase from "to see if…" is unclear.) What process does the "subglacial till conveyor" (line 320-1) refer to? Some work on clarifying the discussion in relation to the specific models would help this section.

Technical comments

12: use of "meter" and "metre" is mixed throughout the text. The rest of the spelling seems to be British English, so suggest changing to "metre" where necessary.

59: "exact mechanism" actually seems a bit vague to me, following from the previous sentences. Perhaps clarify: we need a mechanism to i) generate relief, and ii) account for the observed spatial distribution of that relief (and we can consider particular elements such as size, shape, spacing).

64: add "spacing" to the observed parameters?

105: on first reading of the manuscript, I got a little lost with references to grounding zone, grounding zone cavity, tidal cavity… In your opening sentence you've defined the grounding zone as the zone across which the ice base becomes un/re-grounded with the tides. It would get a bit clunky to add further terms or definitions in that opening sentence, but here (line 105) would be a good place to clarify that this "grounding zone cavity" or "tidal cavity" is the space that opens and closes with each tidal cycle.

133: suggest refer to Fig 2c and 2d.

140: "leaving the till fixed now unyielded" – this is awkward, can you rephrase?

142-4: suggest you insert separate references to Fig 2c and Fig 2d at the relevant points in this sentence.

146: replace "but now" with "which"?

228-9: check the figure panels referred to here – it's not clear to me that 3e/d and 3g/f are the appropriate panels for the statements they are linked to

290: "we do not expect to be able to reproduce ridge asymmetry…" ? What is the purpose of the Graham et al. ref – do they find asymmetry, or not?

291: suggest breaking the paragraph into two, at "Supporting observational evidence…"

325: define these seismic parameters

325: suggest minor rewording of "over a sharp transition" – "over" made me think stratigraphy, then I realise you are more likely referring to an along-flow transition to stiffer till (?)

328: upstream of the grounding zone

334: does "They" refer to Graham et al or Warburton et al?

355, 359: references to Supplementary Material don't match the Part/Table labelling in the supplementary document. Should be Table S2 (line 355) and Text B (359)?

363: put commas around "on the order of a few cm/s" … and therefore (missing e)

407: delete the second "off"

430-2: suggest extend this sentence with something like "but are evident in observational data from Larsen Inlet".

441: can you give this as m/day, as well as km/year (since it serves as a comparison to your results, which you discuss in terms of m/day)?

457: "coarse ridge morphology" is a bit awkward, particularly the word "coarse" which can be used in different ways in referring to landforms and tills. Can you rephrase?

513: misspelling Wåhlin

Supplementary:

You have two captions (one above and one below) for the Supplementary Table in Part A, which is labelled both as Table S1 and S2.

S. 49-50: this sentence duplicates the previous

Unfortunately I couldn't get the videos to play in the Word document. Consider supplying those as separate files.

---

## Author Response (AR1)

General Comments

Hogan et al present a first modelling study aimed at understanding periodic seafloor features (corrugation ridges) that have been imaged proximal to rapidly retreating grounding lines. These features are intriguing and perhaps significant if they can be used as a diagnostic of the speed of retreat. This study provides a context within which these features can now be interpreted. The manuscript is well presented and referenced with ample discussion placing the work in a wider context. An example of the thoroughness of this contribution is quantification of the uncertainty in effective slope introduced by uncertainty in ice thickness gradient close to the grounding zone. This is helpful, as the region of enhanced surface gradient just upstream of grounding zones is often overlooked. In general, the modelling approach is presented as a first effort, and limitations are openly discussed in a useful way. The supplementary material is useful, and the animations provide additional insight.

Specific Comments

These specific comments are minor and intended to help link this study to what we observe at grounding zones.

An omission I noted, which is not a major concern, was the role of englacial sediment delivery, which is then delivered to the grounding zone ocean cavity by subglacial melt, over a length scale partly determined by the debris-rich ice thickness and the ice velocity. I would suggest this mechanism is acknowledged in the introduction, and then need not be addressed further. As it is I believe this mechanism is first mentioned in the Results and Discussion on Line 344. Some explicit statement on the influence that additional accommodation space generated by melt in the grounding zone ocean cavity would also help connect this modelling study to the real system.

*This is an interesting point and we acknowledge that we have followed existing geomorphological studies and focussed on subglacial sediment supply and push-up of that sediment into the subtle corrugation ridges. We did consider including the meltout of debris from debris-rich basal ice as a sediment source (for ridge building) but we realised that the rates of melting and the sediment concentrations in basal ice would likely be far too small than those required to make a ridge every day. We calculated that we need ~0.5 m³ of debris (per m ridge width) to form the ridges as seen at Thwaites Glacier (line 320). Even assuming that a very high basal melt rate (e.g., 30 m/yr at Pine Island Glacier; Shean et al., 2019) was active in the grounding zone this would only equate to 8 cm of basal melt per day. The entire 8-cm thick layer would have to melt (across the 6 m of grounding line retreat) and ALL of that layer would have to be debris (i.e., debris concentration = 100%) to supply 0.5 m³ of material to make a ridge every day. Basal ice debris contents in Antarctica are typically 5-20% (see Christoffersen et al., 2010 and references therein) and, accepting this, basal debris contents are thought to be too small to explain the formation of sedimentary wedge landforms at grounding lines (Anandakrishnan et al., 2007) with the preferred interpretation that sediment is supplied by deformation within the subglacial till layer. Thus, although basal meltout of debris may be important as a sediment source to ice-shelf cavities on longer timescales we do not think it important for the formation of corrugation ridges. Adding to this, the most recent observations from Thwaites grounding zone, made by under-ice robots, show lower-than-expected rates of basal melt and even basal freeze-on close to the grounding line (Davis et al., 2023; Schmidt et al., 2023). Thus, we feel justified to focus on basal sediment delivery to form the corrugation ridges and prefer not to introduce basal melt early in the text as a mechanism for sediment supply, or to create more*

**accommodation space, as neither is entirely applicable to the setting we are modelling. As the reviewer pointed out, we refer to basal melt in the discussion and we are happy to explicitly state that we do not include this sediment source in the methods.**

I also think an upfront statement on the impact of the assumption of the constant 6 m retreat rate is inserted in Materials and Methods (around Line 76, 101) to allay any fears of circularity in the results.

**We had thought that this was clear on line 76 but we see that this may not have been explicit enough. We suggest amending this sentence to include the statement: "thus, a constant retreat rate of 6 m/day is stipulated in our model runs."**

Technical Comments

The manuscript is very well presented and requires very few technical corrections. As a reviewer I appreciate this.

L71 '...due to basal melt..' suggest change to 'due to an increase in basal melt..' as basal melt could be in steady state.

**Yes, we agree and appreciate this small change to make our statement correct.**

L107 expected some mention of debris delivery by basal melt not being addressed around here.

**Thank you for this comment, we can see how an introduction to this process is useful early on in the text when we come back to it later in the discussion. We suggest adding a sentence to clarify that the only sources of sediment considered in our models are from subglacial sediment transport and erosion from the grounding zone bed, and that englacial sediment sources are not modelled.**

L295 ...which perhaps supports dynamic thinning... This is an interesting point that might be lost on first reading. Suggest '...which perhaps supports widespread dynamic thinning...'

**We appreciate this comment to highlight an important point and are happy to adjust the sentence.**

L363 'therefor'

**Corrected spelling here - thank you!**

L407 'lift-off off'

**Corrected by removing the second "off".**

In closing, I thank the authors for their interesting and well-presented study.

*References:*

*Anandakrishnan, S., Catania, G. A., Alley, R. B., and Horgan, H. J.: Discovery of till deposition at the grounding line of Whillans Ice Stream, Science, 315, 1835-1838 (2007).*

*Christoffersen, P., Tulaczyk, S., and Behar, A.: Basal ice sequences in Antarctic ice stream: Exposure of past hydrologic conditions and a principal mode of sediment transfer, J. Geophys. Res., 115, F03034, doi:10.1029/2009JF001430 (2010).*

Davis, P.E.D., Nicholls, K.W., Holland, D.M. et al.: Suppressed basal melting in the eastern Thwaites Glacier grounding zone. Nature **614**, 479–485, https://doi.org/10.1038/s41586-022-05586-0 (2023).

Schmidt, B.E., Washam, P., Davis, P.E.D. et al.: Heterogeneous melting near the Thwaites Glacier grounding line. Nature **614**, 471–478, https://doi.org/10.1038/s41586-022-05691-0 (2023).

Shean, D.E., Joughin, I.R., Dutrieux, P., Smith, B.E., Berthier, E.: Ice shelf basal melt rates from a high-resolution digital elevation model (DEM) record for Pine Island Glacier, Antarctica. The Cryosphere, 13, 2633-2656, https://doi.org/10.5194/tc-13-2633-2019 (2019).

Reviewer 2.

This is a well-constructed manuscript presenting an elegant model–data comparison study into the formation of relatively newly-discovered, delicate grounding line landforms, which have been interpreted as forming daily due to tidal influences on the grounding zone. Verification that tidally modulated grounding line processes – and better understanding of specifically which processes – can produce discrete grounding line landforms is a significant advance that will allow us to better use the palaeo-record to reconstruct past grounding line retreat rates and dynamics and, critically, place bounds on what can be expected in terms of retreat rates from modern ice sheet grounding lines. Here, the different models and their results are well-explained, presented and evaluated; the interpretations and conclusions reached seem sound. (Note that I'm not in a position to comment on the physical formulation of the models.) The limitations of the models (conceptually in set up and in the results they produce) are thoroughly discussed, and assumptions are clearly stated, ensuring that what this work does and does not yet achieve are well-framed. I have a few rather minor comments.

Main comments

1) You state as a given that corrugation ridges are produced at grounding lines. There is good reason for adopting this interpretation for Thwaites and Larsen – occurrence on wedge surfaces, draping MSGLs, displaying lateral continuity – and therefore the approach and research framework here is justified and important. However, other examples have been reported (including by some of the authors) where corrugations are inferred to form in other ways/settings (e.g. not on wedges, and within curvilinear scours attributed to ice shelf or iceberg keels). My reading of the recent Dowdeswell & Graham papers suggests these other interpreted settings are not rejected by new observations, rather that it appears that corrugations might form in various tidally-modulated contexts for re-grounding. I think a couple of additional sentences acknowledging this (for example, around lines 45-52) would be helpful; that the Thwaites and Larsen examples likely formed at the retreating grounding line makes them especially significant and important to understand in terms of formation processes.

***This point is well taken. Our aim was to simply describe the landforms that we are modelling in this study and not to confuse readers with the other types of corrugation ridges not associated with grounding lines. However, this is perhaps slightly disingenuous. We are happy to adjust the text around lines 45-52 to acknowledge these other forms, and we would also like to show a few examples of corrugation ridges from their various settings (grounding lines, in iceberg ploughmarks, from a forward-moving armada of icebergs) as a Supplementary Figure with some explanatory text. Whilst addressing this comment, it occurred to us that some multibeam imagery of corrugation ridges would be a welcome addition to the paper, and we will include this in the Supplementary Figure.***

A follow-on from this distinction is whether the presence of corrugations in other interpreted settings (e.g. iceberg/shelf keels, forward moving) could help us (further) reject/support different mechanisms. Similarity of form and distribution might suggest that a similar mechanism, in some way associated with tidal settling, generates comparable ridge assemblages, whether by a partially grounded sheet or a re-grounding shelf or berg. But not all mechanisms may operate in different settings, or do so with rather different boundary conditions. For example, till flux isn't relevant to iceberg re-grounding cases, which might

support its irrelevance, whereas extrusion (and tidal currents – resuspension?) likely does operate. Can this help your discussion?

***This study (and so our discussion) only considers corrugation ridges formation in grounding line settings so it is difficult to see how this can help our discussion. However, we note that very generally speaking corrugation ridges formed by iceberg keels have larger amplitudes (1-2 m) than those formed in grounding-line settings (<1 m) and we are happy to add a sentence in Section 3.5 "Potential for future modelling work" that this aspect of ridge morphology could be further tested in future models.***

Note that this is also relevant to the conclusion ("only two real-world examples currently known") – yes, from a grounding line, but a little mis-leading regarding the general form.

***We will amend this sentence to read "the only two real-world examples of grounding-line corrugation ridges".***

2) On first reading, I thought that the observational data record is not presented explicitly enough in the main manuscript for the reader to follow which real-world characteristics and observations are involved in the evaluation of the model output. Data-model comparison is the crux of the paper, but the relevant observational data is somewhat hidden in Table 2 and the supplementary material – there is no section in the text that explicitly states "this is what we see in the observational record". I think some minor text adjustments can improve this, both in Section 2.2 where you outline the data, and in Section 3.2 which is set up as where model output is compared to data – I've made specific suggestions below. But I also think an extra panel in Figure 3 and Figure 4 displaying a topographic profile across the Thwaites and Larsen series of corrugations, respectively, would really help. (The figure caption can make it clear what aspects of morphology it is unreasonable to expect models to replicate, in case this is a concern.)

***We are happy to include a brief summary of the observations in the main text (sections 2.2 and 3.2), as suggested, as well as topographic profiles to figures 3 and 4. We were aware that the existing papers on these landforms provide a lot of detail about the morphology of corrugation ridges (e.g. Graham et al., 2013, 2022; Dowdeswell et al., 2020; Batchelor et al., 2020; Davies et al., 2017;Jakobsson et al., 2011) and we did not want to replicate this detail here. But we appreciate the need for a more explicit summary of the observational record for comparison with our model results.***

Text suggestions:

192-198: I think the final sentence here, and thus all the relevant observational data that serves as a basis for evaluation, gets a bit lost. You could lead earlier (e.g. line 192) with a more active statement, to the effect that morphology and spacing observations from the Thwaites data (Graham et al) are given in Table 2, and serve as metrics to be compared against / to evaluate model output. 164 ridges have [… pertinent characteristics…]. You additionally consider morphological elements such as … You could also note, here or at the opening of section 3.2, that while Supplementary Table 1 describes observed morphology as fully as possible, the main traits used for model evaluation are summarised in Table 2.

***We will add more active and explicit statements to section 2.2 to describe the observational data and what aspects of it are used to compare with model results. We will also point to tables 2 and S1 earlier in this section.***

255-262: you could state in this opening paragraph to 3.2 which key observational data characteristics you will use to evaluate model output. Picking through the following sections, it seems to me these include 14-day periodicity, ridge height–spacing correlation, production of individual or composite forms … These are the main characteristics you've reported from model output, but then you also consider (a)symmetry of form, repeatability of form, ridge volume (based on cross-profile height and width)… It would help the reader to allude to these in an introductory way, so we know what to expect from the following analysis, whether just listing in-text as I have done here, or by giving a short summary of what the main characteristics of the observational data actually are.

***In addition to naming the observational data available in section 2.2, we will amend this section to state the observational characteristics that can be compared with the model output.***

Sections 3.2.1, 3.2.2, 3.2.3: these sections are framed as comparing model output to real-world data. 3.2.1 does just this (or begins to), as does the opening paragraph of 3.2.2; 3.2.3 doesn't make such explicit comparisons of ridge morphology or distribution at all, but rather discusses other aspects of the viability of this mechanism. These additional aspects are interesting and important, but beginning each section with a more systematic data-model evaluation would help the overall logical flow and structure of the discussion. Note that even in 3.2.1, you don't directly call upon all your observations to reject the model: state clearly, for example, that observed ridge morphology doesn't match the modelled morphology (composite forms, correlation of height with spacing…) – help the reader work through Table 2, for each modelled mechanism.

***We will go through sections 3.2.1, 3.2.2 and 3.2.3 to ensure each has a logical structure, first stating how the model output matches (or does not match) the observational data and then to discuss the other aspects of each mechanism.***

3) While Section 3.1 reports the output of the model experiments, in the final paragraph (lines 242-253) you move on to a comparison of one particular aspect (production of composite ridges and relationship with slope) with real-world data. Yet you haven't yet introduced any other explicit model-data comparison. This paragraph seems a bit out of place, since the section structure is otherwise clear and logical; the phrase "quite unlike the observations", while you haven't drawn the reader's attention to any observations yet, highlights this section being out of place. I would consider bringing this paragraph into Section 3.2, either up front or towards the end.

***We can see how this paragraph seems out of place in section 3.1; we will move it to section 3.2 as it is a clear implication of the formation of corrugation ridges at grounding lines. Because this paragraph considers both the till extrusion mechanism and the resuspension mechanism we prefer to move this to the end of section 3.2.***

4) It is not clear in your discussion of the till extrusion model (3.2.2) to what extent your discussion refers to extrusion with a flux, or to extrusion with compression. The opening paragraph (line 270-280) seems to suggest that the bed slope effect (composite ridges) should rule out extrusion with a till flux, but that is not explicit as you move through the rest of the section and discuss (a)symmetry, and consistency of form from ridge to ridge. Are you arguing in these cases that extrusion-with-flux and/or extrusion-with-compression are (or aren't) valid? (This is another example of where a more structured or systematic work-through of key traits in Table 2 vs each mechanism – see my second point, above – could help.)

*Thank you for this comment, we acknowledge that the original text did not make this fully explicit. In part this is because, due to the simplicity of our models, the ridges produced by extrusion with the two sediment sources are very similar to one another. We therefore envision 3.2.2 as first describing the morphology produced more generally by including extrusion as a mechanism, followed by a discussion of the size of the observed ridges and the volume of sediment needing to be sourced from either basal sediment flux or GZ compression, and other possible observational markers, as possible ways to distinguish the two mechanisms. We will restructure this section to make this division clearer.*

From line 302 to 322, you then go on to discuss till supply as a "conveyor belt" and other reported estimates of basal sediment flux. In this discussion I no longer follow which modelled mechanism(s) you are using these "flux" estimates and concepts to evaluate: constant flux, extrusion-with-flux, extrusion-with-compression? What does "this" refer to in line 314? (In fact the whole phrase from "to see if…" is unclear.) What process does the "subglacial till conveyor" (line 320-1) refer to? Some work on clarifying the discussion in relation to the specific models would help this section.

*We take the point that this section of the discussion became unclear, here we are considering the basal sediment flux in the extrusion-with-flux mechanism. The till conveyor belt is a concept put forward by Alley et al. in the 1980s for ice flow over a deformable bed and, thus, till transport at the base of an ice sheet.  We call upon the till conveyor as a mechanism to supply the "constant till flux" to the grounding line for our till extrusion model. We will go through the text and clarify which mechanism we are quantifying at each point, and we will use the same terminology throughout to be clear.*

Technical comments

12: use of "meter" and "metre" is mixed throughout the text. The rest of the spelling seems to be British English, so suggest changing to "metre" where necessary.

*We are happy to amend to "metre" throughout the text.*

59: "exact mechanism" actually seems a bit vague to me, following from the previous sentences. Perhaps clarify: we need a mechanism to i) generate relief, and ii) account for the observed spatial distribution of that relief (and we can consider particular elements such as size, shape, spacing).

*We will make this change to the text.*

64: add "spacing" to the observed parameters?

*We will make this change to the text.*

105: on first reading of the manuscript, I got a little lost with references to grounding zone, grounding zone cavity, tidal cavity… In your opening sentence you've defined the grounding zone as the zone across which the ice base becomes un/re-grounded with the tides. It would get a bit clunky to add further terms or definitions in that opening sentence, but here (line 105) would be a good place to clarify that this "grounding zone cavity" or "tidal cavity" is the space that opens and closes with each tidal cycle.

*We will make this addition to the text.*

133: suggest refer to Fig 2c and 2d.

*We will make this change to the text.*

140: "leaving the till fixed now unyielded" – this is awkward, can you rephrase?

*We suggest rephrasing this to: "and the displaced till remains fixed at the low-tide position."*

142-4: suggest you insert separate references to Fig 2c and Fig 2d at the relevant points in this sentence.

*We will add these references.*

146: replace "but now" with "which"?

*We will amend as suggested.*

228-9: check the figure panels referred to here – it's not clear to me that 3e/d and 3g/f are the appropriate panels for the statements they are linked to

*Thank you, we will correct these small errors.*

290: "we do not expect to be able to reproduce ridge asymmetry…" ? What is the purpose of the Graham et al. ref – do they find asymmetry, or not?

*We will remove this reference and amend the following sentence to include that Graham et al. (2022) find that the ridges are symmetric in form, as well as consistent from one ridge to the next.*

291: suggest breaking the paragraph into two, at "Supporting observational evidence…"

*We will make this change.*

325: define these seismic parameters

*We will define the parameters as density, p-wave velocity and s-wave velocity.*

325: suggest minor rewording of "over a sharp transition" – "over" made me think stratigraphy, then I realise you are more likely referring to an along-flow transition to stiffer till (?)

*We suggest changing this to "across a sharp transition".*

328: upstream of the grounding zone

*We have made this change.*

334: does "They" refer to Graham et al or Warburton et al?

*We see this is not clear, the "they" refers to Graham et al. and we have replaced this in the text.*

355, 359: references to Supplementary Material don't match the Part/Table labelling in the supplementary document. Should be Table S2 (line 355) and Text B (359)?

*We have made these changes.*

363: put commas around "on the order of a few cm/s" … and therefore (missing e)

*We have made these revisions.*

407: delete the second "off"

***Amended as suggested.***

430-2: suggest extend this sentence with something like "but are evident in observational data from Larsen Inlet".

***We are happy to amend the sentence to read: "We also observe that both mechanisms produce occasional but discrete double-peaked ridges (see arrows in Figure 4b), as were observed at Larsen Inlet (Dowdeswell et al., 2020; Batchelor et al., 2020), but which were not formed by the models under Thwaites parameters."***

441: can you give this as m/day, as well as km/year (since it serves as a comparison to your results, which you discuss in terms of m/day)?

***We will add that this equates to ~27 m/day, i.e. more than four times the rate inferred for the Thwaites ridges.***

457: "coarse ridge morphology" is a bit awkward, particularly the word "coarse" which can be used in different ways in referring to landforms and tills. Can you rephrase?

***We suggest rephrasing to "…which parameters (surface slope, average till flux, mean retreat rate) control the basic components of ridge morphology, such as amplitude and spacing."***

513: misspelling Wåhlin

***Corrected.***

Supplementary:

You have two captions (one above and one below) for the Supplementary Table in Part A, which is labelled both as Table S1 and S2.

***We prefer to keep the longer caption after the table so we have removed the former (which we saw as a title!) and corrected the table number in the caption to Table S1.***

S. 49-50: this sentence duplicates the previous

***We will remove the duplicated sentence.***

Unfortunately I couldn't get the videos to play in the Word document. Consider supplying those as separate files.

***This is unfortunate! We will supply the videos as separate video files.***